# Facet effect of hematite on the hydrolysis of phthalate esters under ambient humidity conditions

Xin Jin[1], Dingding Wu[1], Cun Liu[2], Shuhan Huang[1], Ziyan Zhou[1], Hao Wu[1], Xiru Chen[1], Meiying Huang[2], Shaoda Zhou[3] & Cheng Gu [1] ✉

Phthalate esters (PAEs) have been extensively used as additives in plastics and wallcovering, causing severe environmental contamination and increasing public health concerns. Here, we find that hematite nanoparticles with specific facet-control can efficiently catalyze PAEs hydrolysis under ambient humidity conditions, with the hydrolysis rates 2 orders of magnitude higher than that in water saturated condition. The catalytic performance of hematite shows a significant facet-dependence with the reactivity in the order {012} > {104} ≫ {001}, related to the atomic array of surface undercoordinated Fe. The {012} and {104} facets with the proper neighboring Fe-Fe distance of 0.34-0.39 nm can bidentately coordinate with PAEs, and thus induce much stronger Lewis-acid catalysis. Our study may inspire the development of nanomaterials with appropriate surface atomic arrays, improves our understanding for the natural transformation of PAEs under low humidity environment, and provides a promising approach to remediate/purify the ambient air contaminated by PAEs.

The usage of phthalate esters (PAEs) as typical plasticizers or additives in commercial products has increased sharply in the past decades[1,2]. While, during aging process, PAEs would be gradually released from plastics and wallcoverings, etc., then leading to widespread environmental contamination[3]. For example, as the extensive usage of agricultural plastic films, the concentration level of PAEs in agricultural soil has reached up to 1232 μg g$^{-1}$ (see ref. [4]). As PAEs are semi-volatile, their contaminations in indoor air (tens to hundreds of ng m$^{-3}$) and in plastic greenhouse (over thousands of ng m$^{-3}$) are attracting more and more attention[5,6]. In China, it was reported that more than 4 million tons of PAEs were emitted into water, air, and soil in the past 60 years[7]. Since PAEs are endocrine disruption compounds[8], understanding their environmental fate is greatly needed. Also, it is demanding for efficient elimination strategies to remove PAEs from the environment.

PAEs are hydrolysable, however, at slow hydrolysis rates with the hydrolysis half-lives ($t_{1/2}$) even up to several years under neutral pH conditions[9]. In our previous studies, we found that chloramphenicol antibiotic could be rapidly hydrolyzed by clay minerals and iron oxides under limited surface moisture conditions, attributing to the strong Brønsted-/Lewis-acidities of the dry mineral surface[10–12]. By contrast, under water-oversaturated conditions, the surface reactions were completely suppressed due to the shielding effect of surface water layers[10–12]. Therefore, moisture plays an important role on surface-mediated reactions. From this point of view, it is prospective that the PAEs adsorbed on mineral surface might proceed more significant hydrolysis under ambient humidity conditions. Considering that PAEs are likely to be associated with soil minerals or mineral dusts[13–15], minerals-mediated hydrolysis might be an important natural attenuation pathway for PAEs in soil and air, where the moisture levels are relatively low. However, these processes are long ignored, and the potential abilities of minerals for catalytic hydrolysis of PAEs are currently unknown.

Hematite can catalyze the hydrolysis of organic contaminants, e.g., antibiotics and organophosphorus esters[12,16]. The catalytic activity of hematite is occasionally dependent on its facet

[1]State Key Laboratory of Pollution Control and Resource Reuse, School of Environment, Nanjing University, 210023 Nanjing, China. [2]Institute of Soil Science, Chinese Academy of Sciences, 210008 Nanjing, China. [3]Nanjing Kaver Scientific Instrument Co. Ltd., 210042 Nanjing, China. ✉e-mail: chenggu@nju.edu.cn

constitution[17]. Although facet-dependent reactivities of oxide minerals have been recognized for decades[18–21], the environmental implications (e.g., contamination removal) are getting more and more attention in recent 10 years, due to the development and feasibility of surface science technologies[22,23]. As large quantities of engineered oxide mineral nanomaterials have been released into the environment[24,25], facet-dependent reactions would play unignorable roles in environmental/geochemical processes[26,27]. For hematite, its facet-dependent reactivity has been widely observed in multiple environmental fields, including iron dissolution[28], (in)organic substances adsorption[29,30], photocatalytic degradation[31–34], and thermo-catalytic oxidation[35], etc. However, almost all these facet-dependent reactions were found in the aqueous phase[17,24–31], and so far, only one study reported the facet-mediated hydrolysis reaction by hematite[17]. Therefore, exploring the hematite facet-mediated hydrolysis of PAEs under ambient humidity conditions can extend our understanding for the environmental fate of PAEs.

To take facet effect into consideration is also on purpose of developing efficient PAEs elimination strategies. PAEs contaminated indoor air would impose direct human exposure, as people spend ~65–90% of their life time in room[36]. Moreover, in China, farmers take an average ~6 h d$^{-1}$ inside plastic greenhouses for agricultural activities[6]. However, there is a lack of promising approach for degrading PAEs in the ambient air. Most of the reported PAEs degradation approaches, involving biodegradation[37], strong base catalyzed hydrolysis[38], radical-based chemical oxidation[39], photocatalysis[40], etc., are all potentially designed for soil remediation or water treatment, while not adequate for air purification. Herein, based on above speculations, the facet-mediated hydrolysis reactivity of hematite might be developed for eliminating PAEs from ambient air. Since studies on oriented synthesis of different facet-constituted hematites have achieved great progress[41,42], engineered facet-controlled hematite nanomaterials could be technically available for environmental applications.

In advance, the hematite facet-orientated mechanisms for PAE hydrolysis should be revealed. It has been well recognized that specific surface geometric and electronic structures would endow different facets with distinct surface properties. For example, the surface site density of the facet-exposed Fe atoms or O atoms usually regulates its adsorption and reaction efficiencies[30,43]. In addition, the surface electronic structure would dominate the catalytic activity in regarding to surface charge state[44], charge transfer potential[31–33], or Lewis acidity[17,30]. However, the known facet-orientated mechanisms might be only the tip of the iceberg, in-depth exploring and understanding new mechanisms would guide the design of more exceedingly active nanomaterials.

In this study, dimethyl phthalate (DMP) and di-$n$-butyl phthalate (DnBP) were chosen as the model PAEs. Three hematite nanoparticles: hematite of nanoplate (HNP, major {001} facet-exposed), nano-rhombohedra (HNR, single {104} facet-exposed), and nano-cube (HNC, single {012} facet-exposed), were synthesized via hydrothermal methods according to the previous reports[31,45,46]. Moreover, their catalytic performances for the hydrolysis of DMP and DnBP were systematically investigated under the relative humidity (RH) of 76%, and the hydrolysis mechanisms were revealed by multiple spectroscopic analyses and theoretical calculations. The results of this study provide insights into the facet effect of hematite and deepen our understanding for the environmental fate of PAEs in soil and air circumstances, meanwhile, supporting soil and air pollution remediation.

## Results and discussion

### Characterizations of the hematite nanoplate (HNP), hematite nano-rhombohedra (HNR), and hematite nano-cube (HNC)

According to the X-ray diffraction (XRD) patterns (Supplementary Fig. 1), the synthesized HNP, HNR and HNC are all assigned as pure hematite ($\alpha$-Fe$_2$O$_3$) with high crystallization. Their morphologies and facet constitutions were identified by scanning electron microscopy (SEM) and high-resolution transmission electron microscopy (HRTEM). The HNP particle is in a uniform shape of hexagonal plate with homogeneous edge length of 80–120 nm and thickness of 14–18 nm (Fig. 1a, d). A close look reflects its lattice fringe of 0.25 nm on the plane and 0.37 nm on edge (Fig. 1g), which is consistent with the {001} and {012} facet, respectively. The HNR particle exhibits a regular rhombohedron shape with edge length of 60–80 nm (Fig. 1b, e). The observed lattice fringe is 0.27 nm, and the dihedral angel between adjacent lateral facets is 62–65° (Fig. 1h), suggesting the single {104} facet. For HNC, it has a pseudo-cubic shape with edge length of ~150 nm (Fig. 1c, f), lattice fringe of 0.37 nm and lattice angle of ~86° (Fig. 1i), indicating the single {012} facet. The facet information for each hematite was further confirmed by the selected area electron diffraction (SAED) pattern (Fig. 1j–l). The particle size distributions of HNC, HNP and HNR are presented in Supplementary Fig. 2. Their isoelectric points were measured as pH 7.0–8.5 (Supplementary Fig. 3), close to the reported values[17,47]. No N 1s signals in X-ray photoelectron spectra (XPS) were detected on HNR and HNC (Supplementary Figs. 5 and 6), suggesting that no organic ligands from synthesis remained on the hematite surfaces. Although the XPS shows clear C 1s signals, the C = O species, which is able to compete for the Lewis-acid sites, comprises only a low amount (9–11% of the total surface carbon content, Supplementary Figs. 4–6). Therefore, the isoelectric points and the XPS results suggest that the HNP, HNR and HNC surfaces are clean with a small amount of adventitious carbon contamination, and the residual carbon is expected to have little influence on the surface reactions. More discussion is provided in Supplementary Note 1. The specific surface areas (SSA) of HNP, HNR and HNC were measured as 24.9, 11.4, and 18.1 m$^2$ g$^{-1}$ by N$_2$-BET method (Supplementary Fig. 7).

### Surface hydrolysis performances

The degradation kinetics of DMP and DnBP by HNP, HNR and HNC under RH 76% are presented in Supplementary Fig. 8. Both DMP and DnBP (2 μmol g$^{-1}$) could be significantly degraded to monomethyl phthalate (MMP) and mono-$n$-butyl phthalate (MnBP), respectively, which are the primary hydrolysis products. The secondary hydrolysis product (i.e., phthalic acid (PA)) was also detected with low yields. All the hydrolysis products were verified by the gas chromatography-mass spectrometer (GC-MS) (Supplementary Fig. 9). The mass balance of the reactants and the products reaches over 90% (Supplementary Fig. 8), suggesting DMP and DnBP are mainly hydrolyzed on hematite surfaces, rather than adsorbed.

The hydrolysis kinetic results of DMP and DnBP could be fitted well to the first-order kinetic model (Fig. 2). Under RH 76%, the surface area normalized hydrolysis rate constants ($k_{BET}$) of DMP and DnBP show significant facet-dependence with the reactivities in the order of HNC {012} > HNR {104} ≫ HNP {001} (Fig. 2a, b). Correspondingly, the hydrolysis half-lives ($t_{1/2}$) are 0.9, 3.0, 14.6 d for DMP, and 1.5, 5.2, 54.2 d for DnBP, on HNC, HNR, HNP, respectively. By contrast, under the water-oversaturated condition (50 mg/L hematite nanoparticles in 200 μL pure water), the order of relative catalytic activities for HNP, HNR and HNC does not change, while, the $k_{BET}$ values for DMP and DnBP are all reduced by ~2-order of magnitude (Fig. 2c, d). The excessive surface water molecules would compete for the reactive sites[10,12], and reduce the Lewis acidity of the exposed Fe (Fig. 3). By the way, the hydrolysis rate of DnBP is much lower than that of DMP on each hematite, ascribing to its longer alkyl chain, which may impose steric-hindrance for either nucleophilic attacking or adsorption[48].

In order to elucidate whether the facet-dependent hydrolysis is specific to PAEs, or is more general, a monoester compound, methyl benzoate (MB), was selected to conduct the same kinetic experiment. As shown in Supplementary Fig. 10, MB could be rapidly hydrolyzed on all three hematites; however, the facet-dependent reactivity for MB in

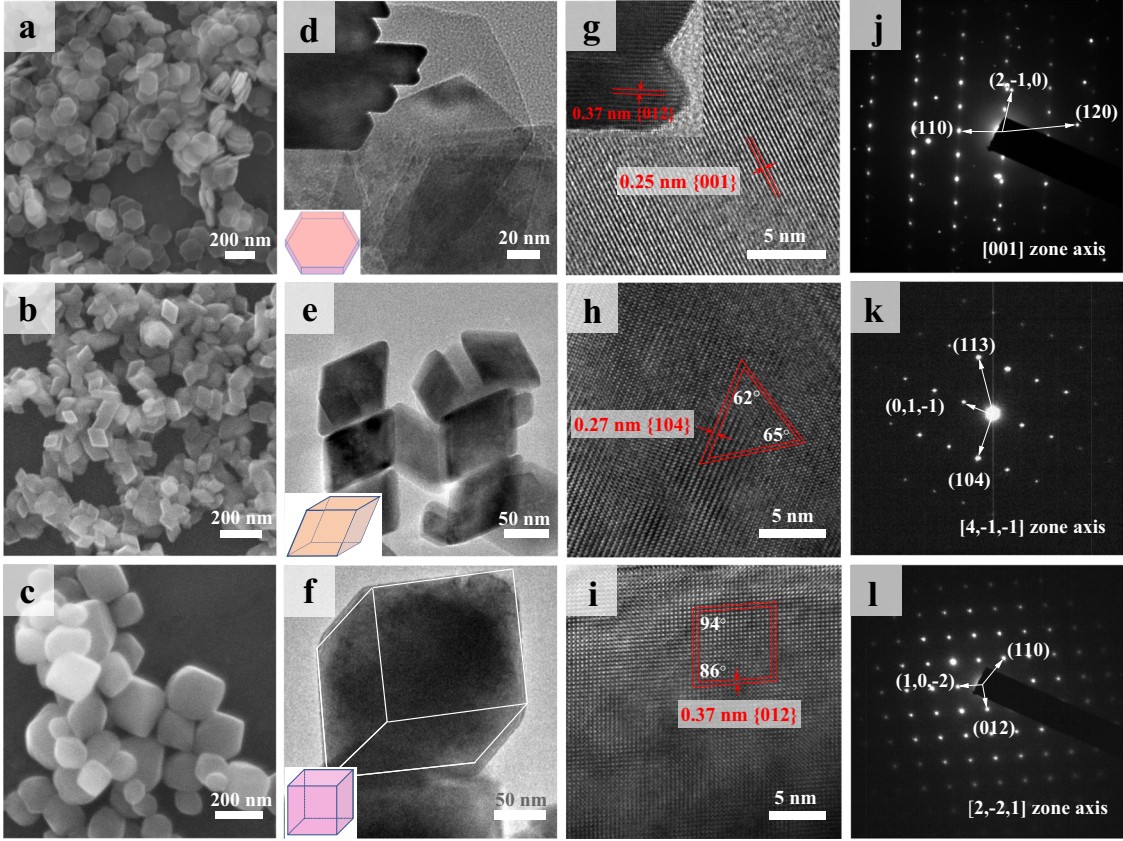

**Fig. 1 | Morphologies and facet constitutions.** SEM images (**a**–**c**), high-magnification TEM images (**d**–**f**), HRTEM images (**g**–**i**) and corresponding SAED patterns (**j**–**l**) of hematite nanoplate (HNP) (**a**, **d**, **g**, **j**), hematite nano-rhombohedra (HNR) (**b**, **e**, **h**, **k**), and hematite nano-cube (HNC) (**c**, **f**, **i**, **l**). The insets in panels **d**–**f** are their geometric models.

the order HNR (0.165 g day$^{-1}$ m$^{-2}$) > HNC (0.116 g day$^{-1}$ m$^{-2}$) > HNP (0.079 g day$^{-1}$ m$^{-2}$) is significantly different from that for DMP/DnBP, suggesting a different mechanism.

**Determination of facet terminations**

The catalytic activity of hematite typically depends on its surface properties. However, the {001}, {104}, {012} facets have multiple theoretical surface terminations (Supplementary Figs. 11–13). To determine the exact termination, we calculated the surface energies of different possible terminations for each facet. It is generally accepted that the facet with the lowest surface energy is thermodynamically favorable. For the {001} facet, after surface relaxation, its layer 1 termination (Fe$_3$–O$_3$–Fe$_6$–R, the subscript represents for the coordination number, Fig. 3a) possesses much lower surface energy ($\gamma = 1.86$ J m$^{-2}$) than the other terminations (Supplementary Table 1), which is in accordance with the previous reports[49–51]. The exposed Fe atoms on {001}-layer 1 termination are three-fold coordinated. However, according to Bader charge analysis, the valence electrons (VEs) remaining on the exposed Fe are relatively low (VEs = 6.39, indicating 1.61 VEs are transferred from Fe to the adjacent O) (Fig. 3a), since the exposed Fe atoms are relaxed inward to the underlayer plane of O atoms with significant charge redistribution.

For the {012} facet, the stoichiometric termination with arm-chair like topology (O$_3$–Fe$_5$–O$_4$–Fe$_6$–O$_4$–R, Fig. 3c) was calculated as the most stable surface ($\gamma = 1.95$ J m$^{-2}$, Supplementary Table 1), which is consistent with the prior studies[52,53]. Since its exposed Fe atoms are fivefold coordinated, the VE number is expectedly low (VE = 6.29, Supplementary Fig. 14d).

Compared to the other two facets, {104} facet was less investigated[54,55]. In this study, the {104}-layer 1 termination

(O$_3$–Fe$_4$–Fe$_4$–O$_4$–O$_4$–R, Fig. 3b) was calculated with the lowest surface energy ($\gamma = 1.94$ J m$^{-2}$), following with the {104}-layer 5 termination (O$_2$–O$_3$–Fe$_5$–Fe$_6$–O$_3$–R, $\gamma = 2.12$ J m$^{-2}$) (Supplementary Fig. 12). As both terminations possess the relatively low $\gamma$, they may co-exist with the domain of {104}-layer 1 as the stable {104} termination. The calculated VEs for the subsurface Fe (6.58) are higher than the outmost Fe (6.36) on the {104}-layer 1 (Fig. 3b).

It is important to note that the higher VEs of Fe cations usually implies lower coordination state and stronger coordination ability. However, when we further applied the diffused reflectance infrared Fourier transform spectroscopy (DRIFTS), using the compound 2-chloro-N,N-dimethylacetamide (Cl-DMA) as the probe molecule, to identify the surface Lewis-acid sites[12], the results are discrepant to the Bader charge analysis (Fig. 3 and Supplementary Fig. 14). For example, HNC exhibits wider red-shift of $\nu_{C=O}$, which denotes to the carbonyl stretching vibration (1621 cm$^{-1}$ on HNC vs. 1628 cm$^{-1}$ on HNP & 1625 cm$^{-1}$ on HNR, Supplementary Fig. 15a–c), suggesting that the HNC surface is more Lewis-acidic. This discrepancy might be ascribed to the existence of surface O-defects on actual HNC surface[53]. By introducing a certain degree of O-defects on the surface of {012} facet model, the Fe atoms adjacent to the O-vacancies possess 6.70–6.74 VEs (Fig. 3c and Supplementary Fig. 14d), corresponding to the stronger Lewis acidity. Meanwhile, a better fit for the IR spectra was also obtained using the {012} facet with O-defect for theoretical modeling (more discussion is shown below). Further study shows that the HNC400 (i.e., HNC calcined at 400 °C for 2 h) exhibits a new shoulder peak at 1582 cm$^{-1}$ (Supplementary Fig. 15f), probably stemming from the thermal desorption of −OH groups and H$_2$O molecules at the defective sites[56], suggesting a small amount of surface defective sites. For comparison, the $\nu_{C=O}$ peaks in

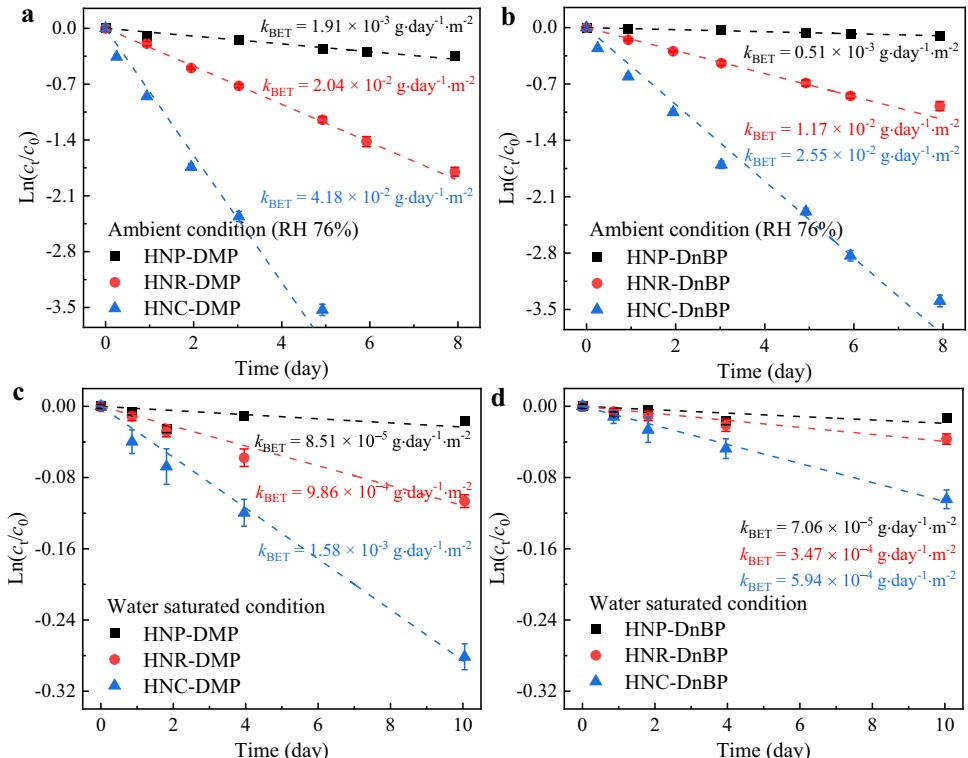

**Fig. 2 | Hydrolysis kinetics.** The fitted first-order kinetics of dimethyl phthalate (DMP) (**a**, **c**) and di-*n*-butyl phthalate (DnBP) (**b**, **d**) on hematite nanoplate (HNP), hematite nano-rhombohedra (HNR), and hematite nano-cube (HNC) under the ambient condition of RH 76% (**a**, **b**), and the water-oversaturated condition (**c**, **d**), respectively. The error bars are expressed as the mean values with standard deviations of two experimental replicates. Source data are provided as a Source Data file.

both HNR and HNR400 samples are narrow and sharp (Supplementary Fig. 15b, e), indicating negligible surface defects on HNR.

Moreover, the surface undercoordinated Fe sites are reactive with water molecules. Under low water partial pressure, $H_2O$ molecules would entirely or dissociatively bond to lattice Fe and O with interactive hydrogen bonding, leading to surface hydroxylation, hydration and protonation (Fig. 3 and Supplementary Fig. 14)[57–60]. The O $1s$ signals in XPS spectra demonstrate that the HNP, HNR and HNC surfaces are partially hydroxylated and hydrated (bond −OH: 16−21.3%, chemisorbed $H_2O$: 13.0−17.3%) even under vacuum condition (Supplementary Figs. 4−6). Therefore, under ambient condition, their surface should be more hydroxylated and hydrated[61]. Although surface hydroxylation/hydration could shield the surface reaction, and affect the valency of the exposed Fe, organic ligands can still compete with the surface bond −OH/$H_2O$ for the undercoordinated Fe sites, especially under low-humidity condition[62], which is strongly evidenced by the desorption of the chemisorbed −OH/$H_2O$ (3000−3750 cm$^{-1}$) substituted by either Cl-DMA or DMP on the three facets (Supplementary Figs. 15 and 16). Therefore, the surface hydroxylation/hydration would not change the coordination mode and the catalytic mechanism. To simplify the facet model, surface hydroxylation/hydration was not involved for modeling. In the current study, the {001}-layer 1, {104}-layer 1 co-existing with layer 5, and the {012}-layer 1 with O-defects are deduced as the surface terminations for HNP, HNR and HNC, respectively (Fig. 3).

## Surface interaction modes

The undercoordinated Fe can offer Lewis-acid complexation sites with the monoester or diester group[16]. The catalytic activity of facet is primarily suspected to correlate to the exposed Fe site density[17,31]. The hydrolysis rates of MB (a monoester compound) on the three facets follow this rule that the ratio of $k_{BET\text{-}\{012\}}$:$k_{BET\text{-}\{104\}}$:$k_{BET\text{-}\{001\}}$ =

1.46:2.09:1.00 is perfectly proportional to the single Fe site density ($D_{si}$) on the three facets, that is $D_{si\text{-}\{012\}}$:$D_{si\text{-}\{104\}}$:$D_{si\text{-}\{001\}}$ = 1.47:2.14:1.00 (Supplementary Table 2). However, the hydrolysis rates of DMP/DnBP on the three facets are not correlated to their $D_{si}$. We propose that the surface coordination modes also significantly regulate the facet effect. For diester compounds, DMP/DnBP are likely to bidentate-coordinate with the neighboring Fe−Fe sites[63].

The feasibility of this hypothesis firstly depends on the molecule geometry of DMP/DnBP, that the two ester groups should orientate to the same side. Although the *trans*-conformation (with their two ester groups on the opposite sides) of DMP or DnBP has relatively lower molecular energy, reconfiguration from *trans*-conformation to *cis*-conformation (with the two ester groups on the same side) is possible, given the low isomerization energies (-6.9 kJ mol$^{-1}$, Supplementary Fig. 17).

Second, the formation of bidentate coordination depends on whether the exposed neighboring Fe sites can sterically match with the double ester groups of DMP/DnBP. The distance between the two carbonyl oxygens of *cis*-DMP/DnBP is -0.34 nm. While, the distances between two neighboring Fe atoms on the {001}, {104} and {012} facets are -0.51, -0.37, and 0.35−0.39 nm, respectively (Fig. 3). It seems that only the {104} and {012} facets can bidentate-coordinate with *cis*-DMP/DnBP, while the {001} facet cannot. Since the experimental concentration of DMP/DnBP (2.0 μmol g$^{-1}$) is far below the saturated extent of adsorption (i.e., >18 μmol g$^{-1}$ for HNC), *cis*-DMP/DnBP has the full probability to form bidentate coordination, endowing the concept of "probability bidentate Fe−Fe site density" ($D_{p\text{-}bi}$). On the {012} facet, each Fe atom neighbors with four Fe atoms (Fig. 3c), resulting in the $D_{p\text{-}bi\text{-}\{012\}}$ of 14.50 sites nm$^{-2}$ (Supplementary Note 2). For the {104} facet, the outmost Fe layer is more accessible for coordination due to the steric accessibility, while the subsurface Fe layer possesses stronger Lewis acidity according to the Bader charge analysis (Fig. 3b).

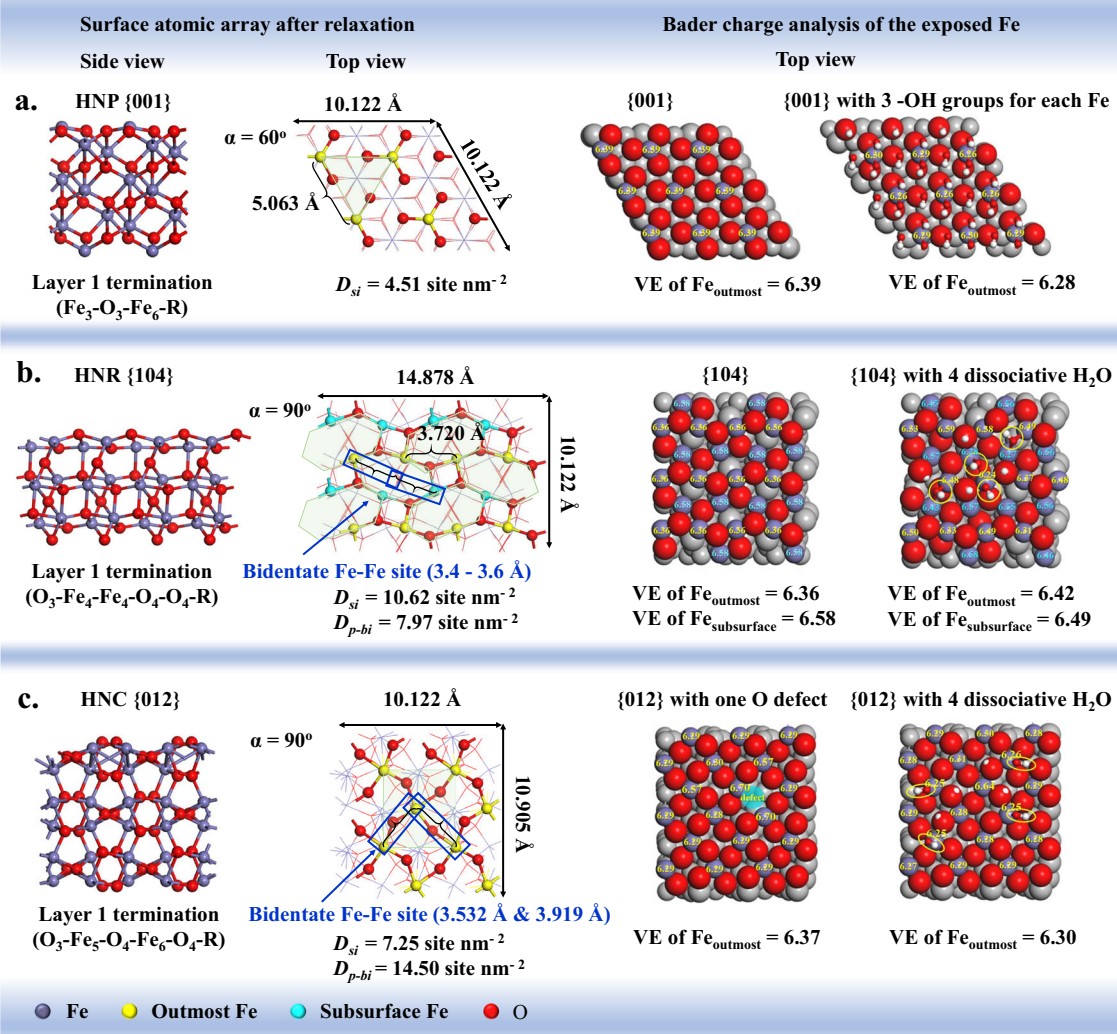

**Fig. 3 | Structural array and Bader charge analysis.** The slab models include (**a**) the {001} facet layer 1 termination of hematite nanoplate (HNP), **b** the {104} facet layer 1 termination of hematite nano-rhombohedra (HNR), and **c** the {012} facet layer 1 termination of hematite nano-cube (HNC). The neighboring Fe–Fe distance and the bidentate Fe–Fe site are labeled. The single Fe site density ($D_{si}$) and the probability bidentate Fe–Fe site density ($D_{p\text{-}bi}$) were calculated according to Supplementary Note 2. The valence electrons (VEs) remaining on the exposed Fe atoms without/with surface hydroxylation are calculated based on Bader charge analysis.

Therefore, the $D_{p\text{-}bi\text{-}\{104\}}$ was calculated as 7.97 sites nm$^{-2}$ (Supplementary Note 2). Interestingly, the $D_{p\text{-}bi\text{-}\{012\}}$:$D_{p\text{-}bi\text{-}\{104\}} = 1.82$ well fits with the hydrolysis rate ratio, $k_{BET\text{-}\{012\}}$:$k_{BET\text{-}104} = 1.6$–2.05 for DMP, and 1.7–2.18 for DnBP (Supplementary Table 2). By contrast, the very low hydrolysis rate of DMP/DnBP on HNP can be ascribed to the failure of forming bidentate coordination on the {001} facet. Therefore, the bidentate coordination mechanism can explain the hydrolysis behaviors of DMP/DnBP on the three facets.

The catalytic activity of the facet also strongly depends on the affinity of the exposed Fe atoms. The in situ DRIFTS measurement (Supplementary Fig. 15) and the Bader charge analysis (Fig. 3 and Supplementary Fig. 14) have implied the surface Lewis-acid properties of the three facets. Herein, we further calculated the turnover numbers (TONs) and turnover frequencies (TOFs) of MB, DMP and DnBP on the three facets, which could reflect the catalytic activity of single Fe site, regardless of the site density[31]. As listed in Supplementary Table 3, the reaction of MB on the three facets shows almost equal TOF/TON values, suggesting the similar Lewis acidity of the single Fe site on the three facets. However, for the reaction of DMP/DnBP, the obtained TON/TOF values are identical on HNR and HNC, but are one order of magnitude higher than that on HNP. This result also provides strong evidence for our hypothesis that the hydrolysis of PAEs on HNR and

HNC is ruled by bidentate coordination mechanism, which could induce much stronger Lewis-acid catalysis.

Subsequently, we simulated the adsorption configurations of DMP using slab models, e.g., the {001}-layer 1, the {104}-layer 1 & layer 5, and the {012}-layer 1 with/without O-defect (Fig. 4a–c and Supplementary Fig. 18). The molecular geometry of DMP is flexible to bidentate-coordinate with the neighboring Fe atoms on the {104} and {012} facets. Notably, although the initial geometry guess anchors *cis*-DMP -2.0 Å above the {104}-layer 1 slab with the two carbonyl groups orientated to the two outmost Fe sites, one of the coordination sites always locates at the bridging position of one outmost Fe and one subsurface Fe (Fig. 4b and Supplementary Fig. 18b, c), as the subsurface Fe possesses stronger Lewis acidity (Fig. 3b). By contrast, the {001} facet can only monodentate-coordinate with one ester group of *trans*-/*cis*-DMP, due to the much wider Fe–Fe distance (Fig. 3a). The calculated adsorption configurations visually confirm the surface interaction modes (Fig. 4).

To obtain further insights into the adsorption mechanism, we applied two methods to calculate the theoretical IR spectra, and compared with the experimental IR spectra.

First, the slab models can represent the periodic hematite surface. So, the obtained DMP-slab adsorption configurations are subjected for

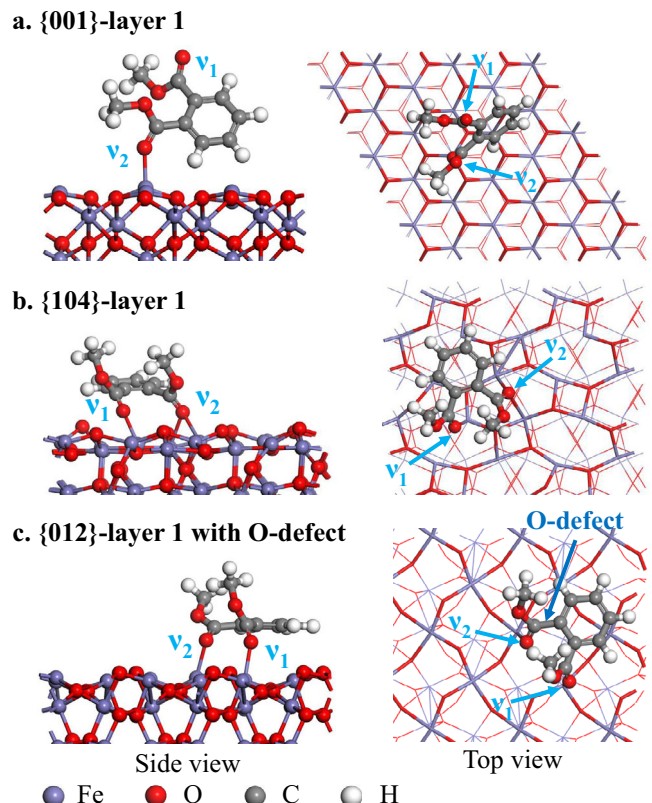

**a. {001}-layer 1**

**b. {104}-layer 1**

**c. {012}-layer 1 with O-defect**

Side view          Top view

● Fe    ● O    ● C    ○ H

**Fig. 4 | Surface adsorption configurations.** The monodentate coordination mode of dimethyl phthalate (DMP) on the slab model of {001}-layer 1 (**a**); the bidentate coordination modes of DMP on the slabs of {104}-layer 1 (**b**), and {012}-layer 1 with O-defects (**c**).

vibration analysis by Vienna ab initio simulation package (VASP) calculation (Fig. 5a). For *tans*-DMP/*cis*-DMP adsorbed on the {001}-layer 1, both show the $v_{C=O}$ at 1708/1700 cm⁻¹ ($v_1$, un-complexed) and 1640/1650 cm⁻¹ ($v_2$, complexed) (Fig. 5a(1) and Supplementary Fig. 18a). For the {104} facet, *cis*-DMP adsorbed on the {104}-layer 1 shows much wider red-shift of the $v_{C=O}$, i.e., at 1614 ($v_1$, complexed by the outmost Fe) and 1587 cm⁻¹ ($v_2$, co-complexed by Fe_subsurface and Fe_outmost) (Fig. 5a(2) and Supplementary Fig. 18b, c). On the other hand, *cis*-DMP adsorbed on the {104}-layer 5 exhibits relatively weaker interaction with the $v_{C=O}$ at 1664 ($v_1$)/1627 ($v_2$) cm⁻¹ (Supplementary Fig. 18d). For the {012} facet without O-defect, the $v_{C=O}$ of *cis*-DMP shows distinct absorption peaks at 1645 ($v_1$)/1625 ($v_2$) cm⁻¹ or 1674 ($v_1$)/1607 ($v_2$) cm⁻¹ (Supplementary Fig. 18e, f). While, on the O-defective site of {012}, the $v_{C=O}$ appears at 1627 ($v_1$) and 1590 ($v_2$) cm⁻¹ (Fig. 5a(3)). Therefore, the theoretical IR spectra of DMP on the slab models directly show the coordination modes. In general, bidentate coordination can induce stronger Lewis-acid interaction than monodentate coordination, imposing a wider red-shift of $v_{C=O}$ even to <1600 nm⁻¹.

Second, the simplified Fe-hydroxyl clusters were also introduced to complex with DMP/DnBP as the representatives of the surface coordination models (Fig. 5b), as the following hydrolysis pathways were calculated using the DMP−Fe-hydroxyl cluster models. This calculation was conducted by the Gaussian software. Taking DMP molecule itself as a benchmark, both the VASP and Gaussian calculations can precisely predict the IR spectrum of pure DMP (Supplementary Fig. 19). In detail, the symmetrical carbonyls of *trans*-DMP possess one theoretical $v_{C=O}$ at ~1730 cm⁻¹ (Fig. 5b(1)). For *cis*-DMP, since its two carbonyls are asymmetric, the $v_{C=O}$ splits to 1724 and 1744 cm⁻¹ (Fig. 5b(2)). When one carbonyl group of *trans*-DMP coordinates with Fe(OH)₃, the un-complexed $v_{C=O}$ locates at 1700 ($v_1$) cm⁻¹, and the

complexed one redshifts to 1645 ($v_2$) cm⁻¹ (Fig. 5b(3)). When both carbonyls of *cis*-DMP bidentate-coordinate with Fe₂O(OH)₄, the $v_{C=O}$ groups exhibit much wider red-shift to 1645 ($v_1$) and 1596 ($v_2$) cm⁻¹ (Fig. 5b(4)). Similar results were obtained for DnBP (Supplementary Fig. 20). The IR spectrum of DMP by Gaussian calculation is in accordance with that by VASP calculation. Therefore, using the simplified Fe-hydroxyl clusters to simulate surface Lewis-acid interaction is also reliable.

The experimental IR spectra of DMP/DnBP adsorbed on HNP, HNR and HNC were investigated by ex situ IR and in situ DRIFTS methods. For the ex situ IR method, DMP/DnBP was initially applied onto the HNP/HNR/HNC nanoparticles, and the mixture was ground to prepared KBr wafers for measurement. As shown in Fig. 5c(2), the two $v_{C=O}$ peaks appear at 1724 cm⁻¹ and 1626 cm⁻¹ on HNP, with the red-shift of ~98 cm⁻¹. Even wider redshifts were observed on HNR (-137 cm⁻¹, Fig. 5c(3)) and HNC (-139 cm⁻¹, Fig. 5c(4)). A similar phenomenon was also observed for DnBP (Supplementary Fig. 21).

By in situ DRIFTS measurement, DMP was blown into the system filled with the HNP/HNR/HNC nanoparticles, then the cumulative adsorption of DMP on the hematite particles was in situ detected. As shown in Fig. 5d, two significant $v_{C=O}$ absorption peaks were observed at ~1718 and 1620 cm⁻¹ on all the three facets. The former indicates the physical adsorbed or weakly bond DMP, and the later should represent for the Lewis-acid coordinated one. After 130 min, two additional shoulder peaks at ~1565 and ~1590 cm⁻¹ start to appear on HNR (Fig. 5d(2) and Supplementary Fig. 22a, b), suggesting bidentate-coordinated adsorption configuration formed on HNR. Such shoulder peaks were also observed on HNC (Fig. 5d(3)), however, did not appear on HNP (Fig. 5d(1)). Both the ex situ and in situ experimental IR spectra are in good accordance with the theoretical IR predictions (Fig. 5), thus, providing the strong evidence that DMP is adsorbed onto HNR and HNC via the bidentate coordination, while via the monodentate coordination on HNP. As indicated by the much wider redshifts of $v_{C=O}$, the bidentate coordination can induce stronger Lewis-acid interaction.

## Hydrolysis pathways of DMP/DnBP by monodentate or bidentate coordination

To gain a better understanding about the surface catalysis mechanism, the hydrolysis pathways of DMP were calculated. For the hydrolysis of pure DMP under neutral pH condition, a two-step hydrolysis process is adopted, which involves two water molecules to form a cyclic transition state (TS) structure (Fig. 6a)[64]. In brief, one $H_2O$ molecule activates the carbonyl O through a hydrogen bonding, simultaneously the second $H_2O$ molecule nucleophilic-attacks the carbonyl C to form the TS1a. As one $H_2O$ molecule leaves, the metastable intermediate (IMa) is formed as the tetrahedral structure. Subsequently, another $H_2O$ molecule participates to induce the proton transfer, following with the ester bond cleavage (TS2a). The activation energies of TS1a and TS2a for the pure DMP are 107.47 and 74.51 kJ mol⁻¹, respectively (Fig. 6c). By contrast, when DMP is bidentate-coordinated with a cluster of Fe₂O(OH₄), a larger cyclic transient structure (TS1b) incorporating two $H_2O$ molecules and one Fe-hydroxyl is formed as the first hydrolysis step (Fig. 6b). Next, the IMb and TS2b are deduced from a similar pathway. The obtained activation energies of TS1b and TS2b are 13.28 and 94.40 kJ mol⁻¹, respectively (Fig. 6d). It verifies that the bidentate coordination of DMP with the undercoordinated Fe atoms can greatly decrease its hydrolysis activation energy, meanwhile, can stabilize the IM structure. Comparing with DMP−Fe₂O(OH₄) and DMP−Fe(OH)₃, the DMP−Fe₂O(OH₄) complex possesses a shorter coordinate bond length (Fe−O: 0.200−0.203 nm, Supplementary Fig. 23b) than that of DMP−Fe(OH)₃ (Fe−O: 0.224 nm, Supplementary Fig. 23a), and the carbonyl C of DMP−Fe₂O(OH₄) is more positively charged (0.903), suggesting that the DMP−Fe₂O(OH₄) complex can proceed more efficient Lewis-acid catalysis. Therefore, nucleophilic hydrolysis is more feasible for the bidentate coordination than the monodentate coordination.

**Fig. 5 | IR spectra comparison between theoretical and experimental ones.**
**a** The theoretical IR spectra of *trans-/cis*-dimethyl phthalate (DMP) on the three
facets by VASP calculation. **b** The theoretical IR spectra of *trans-/cis*-DMP before
and after complexing with Fe(OH)$_3$ or Fe$_2$O(OH)$_4$ cluster, by Gaussian calculation.
The insets are the optimized molecular geometries. **c** The experimental IR spectra
of DMP before and after adsorption onto hematite nanoplate (HNP), hematite

nano-rhombohedra (HNR), and hematite nano-cube (HNC), using ex situ KBr wafer
method. **d** The in situ DRIFTS of DMP adsorbing onto HNP, HNR and HNC under RH
76% as a function of time (0–490 min). The orange shading represents the $\nu_{C=O}$
after coordinating with exposed Fe sites. Source data are provided as a Source
Data file.

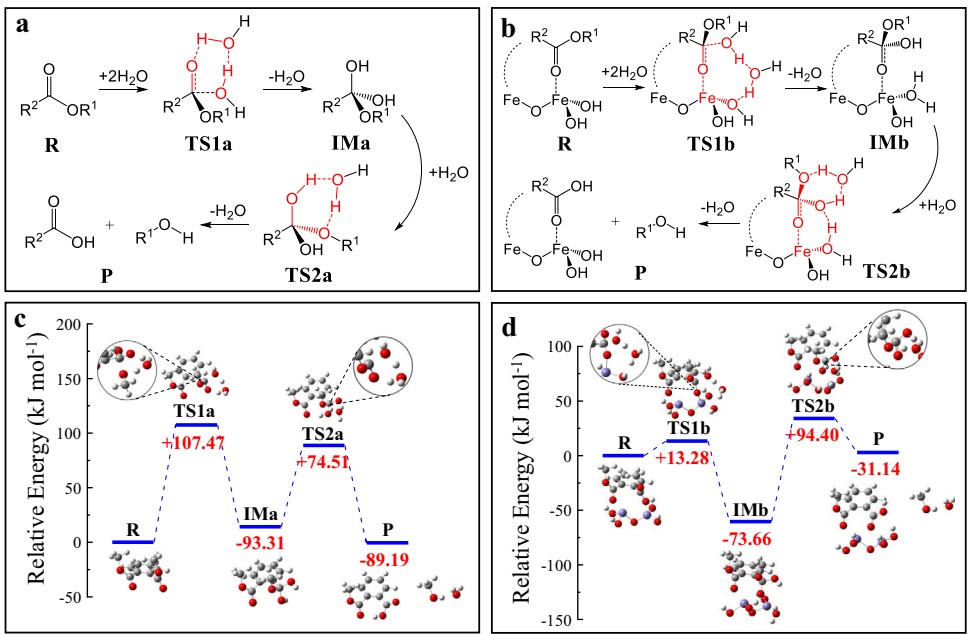

**Fig. 6 | Schematic hydrolysis pathways. a**, **c** The hydrolysis pathway of dimethyl phthalate (DMP) in neutral water by a two-step transition state (TS1 and TS2) mechanism. **b**, **d** The hydrolysis pathway of DMP after bidentate-coordinating with $Fe_2O(OH)_4$ cluster, by a similar two-step transition state mechanism, as the representatives of the surface bidentate coordination forms of DMP on {104} or {012} facet. The acronyms of R, TS, IM, P represent reactant, transition state, intermediate, and products, respectively.

## Passive removal of DMP in the air

Since HNR and HNC have great catalytic hydrolysis activities under ambient conditions, they were subjected for a simple passive treatment experiment to further evaluate their potential applications for degrading air-borne PAEs. According to its vapor pressure (Supplementary Table 4), DMP can evaporate into the headspace with the equilibrium gas phase concentration of $55\,mg\,m^{-3}$. As shown in Supplementary Fig. 24, both HNR and HNC could efficiently adsorb DMP from the air. While, the HNR exhibits much higher adsorption capacity ($36.12\,\mu mol\,g^{-1}$ after 30 d, not yet saturated) than HNC ($18.3\,\mu mol\,g^{-1}$). Both materials could rapidly hydrolyze the adsorbed DMP (with the degradation ratios of 63.7% for HNR and 63.5% for HNC in 30 d) to produce MMP and PA, which are low-volatile, thus can be retained on the hematite surface. A relatively high yield (18%) of PA was observed on HNC. In this experiment, HNC only performs comparable, rather than much higher, hydrolysis rate than HNR. This could be ascribed to the extraordinarily high surface concentration of DMP, as the $D_{p\text{-}di}$ on {012} is more condensed, $D_{p\text{-}di\text{-}\{012\}}$ would overestimate the reaction rate to a greater extent at the high mass loading. Another explanation is that the HNC surface contains a certain degree of defective sites. DMP molecules prefer to interact with Fe atoms at the defective sites on HNC. When the surface DMP concentration exceeds the O-vacancy density, the DMP molecules would react with the remaining fivefold coordinated Fe atoms with weaker Lewis acidity, thereby, the hydrolysis rate is expected to decrease sharply. To further verify this hypothesis, we examined the concentration effect ($[DMP]_0 = 2\text{--}20\,\mu mol\,g^{-1}$) on the reaction. As shown in Supplementary Fig. 25, the hydrolysis rates of DMP on HNC decrease greatly as the initial DMP concentration exceeds $2\,\mu mol\,g^{-1}$, suggesting that the O-vacancy density on HNC surface might be $\sim 2\,\mu mol\,g^{-1}$ (i.e., $0.07\,sites\,nm^{-2}$), accounting for $\sim 2\%$ of the total surface area. By comparison, the hydrolysis rates of DMP on HNR only slightly decreased in the applied concentration range, due to its stoichiometric surface condition.

However, by considering that the PAE concentration in actual contaminated air (in ppt level) is ~6 orders of magnitude lower than that used in this experiment (in ppm level)[5,6], the surface accumulated PAEs would not be in this level. Assuming that the PAE concentration in the air is $10\text{--}1000\,ng\,m^{-3}$, the estimated treatment capacity for HNC could be over $3500\,m^3$ (air) $g^{-1}$, which is adequate for practical application. At low mass loading, the HNC has the advantage due to a faster hydrolysis rate, and more thorough degradation. By contrast, the HNR nanoparticles are more applicable for the heavily contaminated scenarios.

In summary, our study demonstrates the specific facet effect of hematite on mediating the hydrolysis of PAEs. The reaction efficiency under ambient humidity conditions is much higher than that in water-oversaturated condition. The reaction exhibits a new facet-dependent mechanism, that the hematite facet with the exposed neighboring Fe atoms of $0.34\text{--}0.39\,nm$ can bidentate-coordinate with PAEs, thus providing stronger Lewis-acid interaction. This mechanism would inspire researchers to develop nanomaterials with ideal surface atomic (i.e., the active sites) array to achieve higher catalytic performance. For example, we also synthesized the {101} and {001} facet-controlled $TiO_2$ nanoparticles for the same hydrolysis experiment (Supplementary Fig. 26a, d)[65]. Meanwhile, one commercial amorphous $TiO_2$ nanopowder was used for comparison. The neighboring Ti–Ti distances on both {101} and {001} facets are $0.38\,nm$ (Supplementary Fig. 26c, f), which is appropriate for bidentate coordination with DMP, according to our proposed mechanism. As shown in Supplementary Fig. 26g, both $TiO_2$-{101} and $TiO_2$-{001} perform considerable (apparent) hydrolysis reactivity for DMP, even more reactive than HNR. However, the catalytic activity of the amorphous $TiO_2$ nanopowder is much lower, suggesting that the plane facet and bidentate coordination are the two important prerequisites for rapid surface-catalyzed hydrolysis reactions. While more investigations are still needed to solidify this mechanism. Herein, our study adds new understanding for the natural transformation pathway of PAEs in low-moist environment (e.g., upland soils, ambient air, etc.), and provides a new approach for purifying PAEs contamination in the air.

## Methods
### Chemicals and reagents
This information is provided in Supplementary Method 1.

### Synthesis and characterization of the hematite nanoparticles

We synthesized three types of hematite nanoparticles with specific facets (HNP, HNR and HNC) via hydrothermal methods (Supplementary Method 2)[17,38,39]. The purity, morphology, facet constitution, particle size distribution, surface zeta potential, XPS-surface analysis, SSA of the synthesized HNP, HNR and HNC were characterized according to Supplementary Method 3.

### Kinetic experiment

The hydrolysis experiments were conducted in desiccators with a constant RH of 76%, which was controlled by the saturated NaCl solution at the bottom of the desiccator. Before each kinetic experiment, the hematite nanoparticles were oven-dried (105 °C) for overnight to remove excessive surface-free water. Then, 200 μL the 0.5 mM acetone-based DMP or DnBP solution was applied onto 50 mg HNP, HNR or HNC nanoparticles in a glass vial, to obtain the initial DMP/DnBP concentration of 2.0 μmol g$^{-1}$. Once the acetone was entirely evaporated, the vial was transferred into the desiccator for reaction. A parallel group was set by adding 200 μL pure water into the vial after acetone evaporation, to have the water-oversaturated condition, and the vial was then covered. During the reaction, at certain time interval, the residual DMP/DnBP and the hydrolysis products were extracted by the addition of 5 mL extraction agent (50% methanol in water (vol.%) with 4 mM ethylenediaminetetraacetic acid disodium (EDTA), pH = 10). The extractant was then filtered and acidified for high-performance liquid chromatography (HPLC) analysis (Supplementary Method 4). The hydrolysis products were further identified by gas chromatography-mass spectrometer (GC-MS, Thermo Scientific, USA) (Supplementary Method 5).

To evaluate the potential applications of HNR and HNC for air purification, a passive treatment experiment was conducted in the desiccator at RH 76% and 35 °C. The pure DMP (in liquid) was filled into an open vial inside the desiccator, which was surrounded by vials containing 50 mg HNR or HNC nanoparticles. DMP was evaporated into the headspace, then adsorbed and simultaneously degraded on HNR or HNC. The residual DMP and the degradation products on the hematite surface were extracted and measured as described above.

### Infrared (IR) spectroscopic analysis

The interaction of DMP/DnBP with each facet was identified on an IR spectrometer with a liquid-nitrogen-cooled mercury cadmium telluride detector (Bruker Tensor 27, Bruker, Germany). The acetone-based DMP or DnBP was inoculated into each hematite (20 μmol g$^{-1}$), then the mixture was ground and prepared into KBr wafer (1%, wt. %) for measurement. In addition, by DRIFTS measurement, each hematite nanoparticles were filled into an HVC-DRP-5 accessory (Harrick Scientific, USA). DMP was carried by a N$_2$ flow (15 mL min$^{-1}$) in combination with a second channel of the humidified N$_2$ flow (90 ml L$^{-1}$, RH 76%) to pass through the hematite powder. As gaseous DMP was adsorbed by the hematite nanoparticles, its IR signal was recorded in situ as a function of time. The DRITFS system was operated at 25 °C. To indicate the surface Lewis-acid site distribution, the same DRIFTS method was applied using Cl-DMA as the probe molecule. This method was described in our previous study[12].

### Density functional theory (DFT) calculation

DFT calculations were performed to obtain the surface energies, the adsorption configurations, and the corresponding theoretical IR spectra, using the VASP 5.4 software. The generalized gradient approximation (GGA) approach with the Perdew–Bruke–Ernzenrhof (PBE) exchange-correlation functional, and a cutoff energy of 550 eV for the plane-wave basis set were used for all the calculations. DFT + $U$ method was adopted to treat the 3d orbital electrons of Fe with $U_{eff} = 5.00$ eV[51]. Core electrons are represented by the Project Augmented Wave (PAW) pseudopotentials with 8 and 6 valance electrons

for Fe and O, respectively. The harmonic frequencies were calculated after the determination of the Hessian matrix, while vibration intensities were evaluated through DFPT linear response calculations using Born effective charges, using the formula by Gianozzi and Baroni[66,67]. The IR spectrum was further plotted by the Multiwfn software[68].

Simplified models of DMP complexing with Fe(OH)$_3$ and Fe$_2$O(OH)$_4$ clusters were adopted as the representatives of the surface Lewis-acid interaction. DFT calculations of IR spectra and hydrolysis pathway simulations were performed by the Gaussian 09 program[69], using the B3LYP/6-311 g(d,p) method for C, H, O, and the LANL2DZ basis set for Fe, in addition with the SMD solvent model. The hydrolysis-activated energies were obtained by searching the hydrolysis transition states (TS). To improve the computational accuracy, the obtained geometries of ground state and TS were further employed for single-point energy calculation using the wB97M-v/ def2tzvp/SMD method by ORCA 5.0[70].

## Data availability

The data supporting the findings of this study are available from the corresponding author on request. Source data are provided with this paper.

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

## Acknowledgements

This work is funded by the National Natural Science Foundation of China (21777066, 41703090, and 22176092), the Natural Science Foundation of Jiangsu Province of China (BK20211154), the Fundamental Research Funds for the Central Universities (2022300311). We sincerely thank Prof. Lan Ling (College of Environmental Science and Engineering, Tongji University) and Dr. Yinping Zhang (Analytical & Testing Center, Nanjing Normal University) for their assistance in sample analysis. We are grateful to the High-Performance Computing Center of Nanjing University for using the computation resources.

## Author contributions

X.J. and C.G. conceived the idea for the study. D.W. and S.H. conducted the kinetic experiments. D.W., Z.Z., and H.W. synthesized and characterized the hematite nanoparticles. X.J., D.W., C.L., and M.H performed the theoretical calculations. D.W. and X.C. contributed to the identification of the products. S.Z. designed the in situ IR equipment. X.J., D.W., and C.G. discussed the results and wrote the manuscript with input from all co-authors.

## Competing interests

The authors declare no competing interests.
