## [Peer Review File · Nature Communications]

REVIEWER COMMENTS

Reviewer #1 (Remarks to the Author):

The authors of submitted manuscript investigated the transformation of PAEs by hematite. Dimethyl phthalate (DMP) and D-n-butyl phthalate (DnBP) were chosen as model PAEs. The authors described the influence of different hematite facets on catalytic transformation of PAEs. Spectroscopic evidences are found for the formation of bidentate coordination mode at the hematite surface.

This investigation is more appropriate for some international journal covering the field of environmental protection.

I do not recommend this manuscript as publication in Nature communications

Reviewer #2 (Remarks to the Author):

Wu et al. investigated the facet-dependent hydrolysis of phthalates over hematite nanoparticles. The authors prepared three well-defined nanoparticles with predominantly exposed {001}, {104}, and {012} facets using hydrothermal methods. They demonstrated that the catalytic performance of hematite showed a significant facet-governing reactivity in the order {012} > {104} >> {001}. They used different methods to explain the facet-dependent mechanism, such as IR and cluster theoretical modeling. They concluded that proper neighboring Fe-Fe distance of 0.37 nm favored the formation of bidentate-coordinate with PAEs and thus promoted PAEs hydrolysis. The facet-dependent reactivity is different from the traditional hydrolysis mechanism, providing new insights into the structure-activity relationships. The manuscript is well organized, and the results support the conclusion, although there are some issues. Overall, the manuscript meets the scope of the journal. In my opinion, the manuscript can be published if the authors can make a thorough revision. The specific questions and concerns are summarized below.

Major

1. How did the authors confirm the atomic arrangements of these exposed surfaces? Many papers discuss the surface atomic arrangements for hematite, which is complicated. For instance, the hematite {001} has three atomic terminations: O termination, one layer Fe termination, and double-layer Fe termination. The density of the undercoordinated Fe atoms should be different for different atomic terminations. However, such important information was not included and discussed in the manuscript.

2. As the reactivity is correlated to the surface with active sites, the reviewer would suggest that the turnover frequency (TOF) of the catalysts based on each Fe active site and turn number (TON) of the contaminants need to be compared for the three surfaces.

3. The authors used some organics (or additives) to prepare the hematite nanoparticles. Whether the organic ligands are completely removed needs to be confirmed. The contaminated organic carbon might influence the interfacial structures of phthalates complexed on these surfaces due to the possible competitive adsorption. The C 1s XPS should be feasible to prove whether the hematite surface was contaminated with carbon. What is more, how to wash the samples needs to be provided.

4. The authors used iron cluster models to simulate the iron-DMP/DnBP complexes; however, some problems exist. The actual surface of hematite is periodic, so clusters cannot represent the periodic surface. The experimental and theoretical IR did not match well with each well; the rational use of the cluster modeling needs to be clarified.

Minor

1. Are the configurations in figure 3a were checked by DFT calculation?

2. The size distribution of these nanoparticles may be needed because sizes seem not well distributed from the SEM images, especially for the pseudo-cube nanoparticles.

3. Some related papers may be cited because they are conceptually related on a higher level, such as Facet-dependent contaminant removal properties of hematite nanocrystals and their environmental implications. *Environ. Sci.: Nano* 2018, 5 (8), 1790-1806.

4. The reactivity of the nanoparticles needs to be compared to other metal oxides for the hydrolysis of phthalates.

5. The iron atoms between two adjacent surfaces are also undercoordinated, and thus the contribution of the reactivity needs to be evaluated.

6. Line 117, dihedral angle with the adjacent lateral facet was 86°, not 90°. The nanocube nanoparticle is a pseudo-cube.

7. Figure 4, Please indicate what the atoms are in different colors.

8. TOC art is not fully displayed.

Reviewer #3 (Remarks to the Author):

The submitted work deals with the role of hematite crystals on the hydrolysis of phthalates in vapour phase, and especially the difference between the face orientations.

The title claims the “non-aqueous” hydrolysis, but the authors mean that the reaction take place in presence of hydrated vapour (relative humidity of 76%). So, the title should be changed into “(water) vapour phase” rather than “non-aqueous”.

The concept of face-dependent reactivity has existed for decades, as in the earlier work by Knozinger in catalysis (doi 10.1080/03602457808080878 for example in 1978), or Hiemstra in environmental sciences (doi 10.1016/0021-9797(89)90284-1 in 1989). Unfortunately, the bibliography of the submitted work only refer to the articles of 2000s (with very few exceptions). A few lines about the history of this approach would be welcomed.

Detailed comments

Li. 106. The orientation of faces has been characterized by microscopy. The surface reactivity depends on the purity of these surfaces, so a low amount of adsorbed ions/molecules can change it. It would be interesting to measure the isoelectric point of this solids by zetametry, and compares them to calculated ones to ensure that the difference in reactivity comes from the orientation and not from the presence of surface residual chemicals used in the synthesis.

Li. 115. Has the presence of defects been investigated? They are expected to have a huge effect on reactivity.

Li. 121. The number of digits for the SSAs should be decreased (18.1 instead of 18.08 for example).

Li 141. The term “water content of 400%” is troublesome, as “over-wet condition”. If the authors mean that liquid water is present, they could give the calculated value of water layers at the surface of the particles for example.

Li. 165. Uncoordinated Fe sites are expected to be very reactive towards water molecules as a vapour or in a liquid. Thus, this type of sites may be not exist in the conditions of the synthesis (in water or at RH > 0%). This type of question arises in the surface chemistry of transition aluminas, and

the authors should give some evidence (experimental or from calculations) that the terminations they propose (li.161-163) actually exist in their systems.

TOC. The graph seems cut. Moreover, there are too many details, and it is difficult to understand in an independent way.

Fig. 4. "neutral"

Reviewer #1:

The authors of submitted manuscript investigated the transformation of PAEs by hematite. Dimethyl phthalate (DMP) and D-n-butyl phthalate (DnBP) were chosen as model PAEs. The authors described the influence of different hematite facets on catalytic transformation of PAEs. Spectroscopic evidences are found for the formation of bidentate coordination mode at the hematite surface.

This investigation is more appropriate for some international journal covering the field of environmental protection.

I do not recommend this manuscript as publication in Nature communications.

Response: Thank you for the comments even though the reviewer did not recommend our paper as publication in *Nature Communications*. Herein, we would like to briefly explain the novelties of this manuscript. In this study, we investigated the specific facet effect of hematite on the hydrolysis of PAEs under ambient humidity/moisture condition, and found that the hematite {104} and hematite {012} exhibited 1-2 orders of magnitude higher catalytic activities than the hematite {001}. We proposed that the neighboring Fe atoms on the facet with a proper distance of 0.35-0.40 nm could form the bidentate coordination complex with PAEs, then inducing stronger Lewis-acid catalytic degradation. We think the novelties of this study stand on the following three aspects:

1) It is the first report to investigate the facet selectivity of minerals for “non-aqueous reaction” of organic contaminants. Currently, there are abundant studies about the mineral facet-dependent environmental processes, e.g., iron dissolution (Huang et al., 2017), adsorption of (in)organic substances (Huang et al., 2016; Shen et al., 2020), photocatalytic degradation (Guo et al., 2021; Huang et al., 2019; Zong et al., 2021; Zhou et al., 2012;), thermo-catalytic oxidation (Liu et al., 2011), and surface hydrolysis reaction (Li et al., 2020), etc. However, almost all these facet-dependent reactions were conducted in aqueous solution. To the best of our knowledge, there are still no reports about facet-selective non-aqueous transformation of organic

contaminants. Although this process may be important in soil and air environments, as soil is only partially hydrated in most cases, and mineral particles in the air are exposed under ambient humidity, this transformation pathway is usually ignored. In this study, our results reveal that the relatively dry mineral surface with certain facet can perform much higher catalytic reactivity under ambient humidity/moisture conditions compared to the reaction occurring in solution.

2) The proposed bidentate binuclear complexation mechanism is new. The facet reactivity usually depends on the abundance and affinity of surface exposed reactive sites (Mei et al., 2020; Shen et al., 2020). While, our study reveals that the surface Fe atomic array is also important for certain reactions. It inspires researchers to select and design novel nanomaterials with proper surface atomic array (i.e., the active sites) to achieve high catalytic reactivity. For example, besides hematite, we also synthesized the {001} and {101} facet dominated titanium oxide (TiO₂) nanoparticles (Li et al., 2015). Since the neighboring Ti-Ti site distance on both {001} and {101} facets is ideal for bidentate coordination (i.e., 0.38 nm), both minerals perform considerable hydrolysis rates for dimethyl phthalate (DMP) under RH 76% (Figure R1). While, the commercial amorphous TiO₂ nanopowder exhibits poor catalytic activity due to its amorphous surface condition (Figure R1). The results further emphasize the import roles of plane facet and bidentate-coordination for the hydrolysis reaction.

3) The facet dependent hydrolysis of PAEs is applicable for actual environmental remediation/purification. In this study, we also demonstrate that the facet specific hematite exhibits promising applicability for purifying PAE contamination in plastic greenhouse and indoor air, which is still lack of proper purification strategies up to date. Currently, most of the reported PAE degradation approaches, involving biodegradation (Di Gennaro et al., 2005), strong base catalyzed hydrolysis (Xu et al., 2010), radical based chemical oxidation (Yao et al., 2020), photocatalysis (Cai et al., 2021), *etc.*, are designed for soil remediation or water treatment, while not adequate for air purification.

Based on the three justifications, we believe our study is scientifically valuable.

On the other hand, “*Nature Communications*” is a multidisciplinary journal covering all areas of biological, health, physical, chemical and earth sciences. We think our manuscript falls within the scope of *Nature Communications*, subjecting to the discipline of Earth and Environmental Science/Environmental Sciences/Environmental Chemistry. We carefully searched the published articles in *Nature Communications* within the scope “*environmental chemistry + environmental protection + organic pollutant degradation*”, and found several articles since 2021, as listed below. It indicates that *Nature Communications* also welcomes investigations in the field of environmental protection.

- (1) Li, J., Li, X., Da, Y. et al. *Sustainable environmental remediation via biomimetic multifunctional lignocellulosic nano-framework*. *Nat. Commun.* 13, 4368 (2022). <https://doi.org/10.1038/s41467-022-31881-5>. This study designed a new biomimetic multifunctional lignocellulosic nano-framework from chemically modified lignocellulosic biomass. The new material can efficiently adsorb PFAS, and can provide colony for fungus. Such that, the adsorbed PFASs can be microbial degraded in-situ.
- (2) Zhang, Y., Huang, G., Winter, L. et al. *Simultaneous nanocatalytic surface activation of pollutants and oxidants for highly efficient water decontamination*. *Nat. Commun.* 13, 3005 (2022). <https://doi.org/10.1038/s41467-022-30560-9>. This study reported a new advanced oxidation mechanism, which occurred on the nanocatalyst surface, with the characteristics of synchronous degradation and mineralization.
- (3) Makam, P., Yamijala, S.S.R.K.C., Bhadram, V.S. et al. *Single amino acid bionanozyme for environmental remediation*. *Nat. Commun.* 13, 1505 (2022). <https://doi.org/10.1038/s41467-022-28942-0>. This study synthesized a single amino acid bionanozyme, which can catalyze the rapid oxidation of phenolic contaminants.
- (4) He, F., Weon, S., Jeon, W. et al. *Self-wetting triphase photocatalysis for effective and selective removal of hydrophilic volatile organic compounds in air*. *Nat.*

- Commun.* 12, 6259 (2021). <https://doi.org/10.1038/s41467-021-26541-z>. This research introduced a novel method for utilizing periodic acid (PA), which is a highly hygroscopic substance, to create an *in-situ* water layer on a photocatalytic material WO₃ under ambient condition, and to selectively and efficiently catalyze the photo-transformation of some hydrophilic volatile organic compounds (VOCs) (e.g., acetaldehyde and formaldehyde).
- (5) *Chu, C., Huang, D., Gupta, S. et al. Neighboring Pd single atoms surpass isolated single atoms for selective hydrodehalogenation catalysis. Nat. Commun.* 12, 5179 (2021). <https://doi.org/10.1038/s41467-021-25526-2>. This research reported that the neighboring Pd single atoms can contribute to a superior performance for selective hydrodehalogenation of organohalides (e.g., 4-chlorophenol), and the DFT calculation also supports that the two neighboring Pd atoms are necessary for C-Cl bond cleavage.
- (6) *Wang, Y., Xu, Y., Dong, S. et al. Ultrasonic activation of inert poly(tetrafluoroethylene) enables piezocatalytic generation of reactive oxygen species. Nat. Commun.* 12, 3508 (2021). <https://doi.org/10.1038/s41467-021-23921-3>. This research reported the PTFE particles as a new piezo-catalytic material for environmental degradation of organic pollutants (e.g., acid orange 7, methylene blue, Nitrobenzene, 4-chlorophenol) under ultrasonic activation. The generation of reactive oxygen species (ROS) by piezocatalytic mechanism should be responsible for the high performance.
- (7) *Min, Y., Zhou, X., Chen, JJ. et al. Integrating single-cobalt-site and electric field of boron nitride in dechlorination electrocatalysts by bioinspired design. Nat. Commun.* 12, 303 (2021). <https://doi.org/10.1038/s41467-020-20619-w>. This article reported a single-atomic-site Co catalyst supported by carbon doped boron nitride for electro-catalytic reduction of chloramphenicol, which is a kind of halogen-containing antibiotics.

Figure R1. The scanning electron microscopy (SEM) images of the synthesized TiO₂-{001} (a) and TiO₂-{101} nanoparticles (d). The geometric structures of TiO₂-{001} (b, c) and TiO₂-{101} (e, f) from side view (b, e) and top view (c, f). The catalytic performance of TiO₂-{001} and TiO₂-{101} for the hydrolysis of DMP (2.0 μmol·g⁻¹) under RH 76%, compared to the performance of commercial amorphous TiO₂ nanopowder (g).

References:

- Cai, J., Niu, B., Zhao, H. and Zhao, G. 2021. Selective photoelectrocatalytic removal for group-targets of phthalic esters. *Environmental Science & Technology* 55(4), 2618-2627.
- Di Gennaro, P., Collina, E., Franzetti, A., Lasagni, M., Luridiana, A., Pitea, D. and Bestetti, G. 2005. Bioremediation of diethylhexyl phthalate contaminated soil: A feasibility study in slurry- and solid-phase reactors. *Environmental Science & Technology* 39(1), 325-330.
- Guo, T., Jiang, L., Wang, K., Li, Y., Huang, H., Wu, X. and Zhang, G. 2021. Efficient persulfate activation by hematite nanocrystals for degradation of organic pollutants under visible light irradiation: Facet-dependent catalytic performance and degradation mechanism. *Applied Catalysis B: Environmental* 286, 119883.
- Huang, X., Hou, X., Song, F., Zhao, J. and Zhang, L. 2016. Facet-dependent Cr(VI) adsorption of hematite nanocrystals. *Environmental Science & Technology* 50(4), 1964-1972.
- Huang, X., Hou, X., Song, F., Zhao, J. & Zhang, L. 2017. Ascorbate Induced Facet Dependent Reductive Dissolution of Hematite Nanocrystals. *Journal of Physical Chemistry C* 121, 1121.
- Huang, X., Chen, Y., Walter, E., Zong, M., Wang, Y., Zhang, X., Qafoku, O., Wang, Z. and Rosso, K.M. 2019. Facet-specific photocatalytic degradation of organics by heterogeneous fenton chemistry on hematite nanoparticles. *Environmental Science & Technology* 53(17), 10197-10207.
- Li, C., Koenigsmann, C., Ding, W., Rudshteyn, B., Yang, K.R., Regan, K.P., Konezny, S.J., Batista, V.S., Brudvig, G.W., Schmuttenmaer, C.A. and Kim, J.H. 2015. Facet-dependent photoelectrochemical performance of TiO₂ nanostructures: An experimental and computational study. *Journal of the American Chemical Society* 137(4), 1520-1529.
- Li, T., Zhong, W., Jing, C., Li, X., Zhang, T., Jiang, C. and Chen, W. 2020. Enhanced hydrolysis of p-nitrophenyl phosphate by iron (hydr)oxide nanoparticles: Roles of exposed facets. *Environmental Science & Technology* 54(14), 8658-8667.
- Liu, X., Liu, J., Chang, Z., Sun, X. and Li, Y. 2011. Crystal plane effect of Fe₂O₃ with various morphologies on CO catalytic oxidation. *Catalysis Communications* 12(6), 530-534.
- Mei, H., Liu, Y., Tan, X., Feng, J., Ai, Y. and Fang, M. 2020. U(VI) adsorption on hematite nanocrystals: Insights into the reactivity of {001} and {012} facets. *Journal of Hazardous Materials* 399, 123028.
- Shen, Z., Zhang, Z., Li, T., Yao, Q., Zhang, T. and Chen, W. 2020. Facet-dependent adsorption and fractionation of natural organic matter on crystalline metal oxide nanoparticles. *Environmental Science & Technology* 54(14), 8622-8631.
- Xu, Z., Zhang, W., Lv, L., Pan, B., Lan, P. and Zhang, Q. 2010. A new approach to catalytic degradation of dimethyl phthalate by a macroporous OH-type strongly basic anion exchange resin. *Environmental Science & Technology* 44(8), 3130-3135.
- Yao, J., Yu, Y., Qu, R., Chen, J., Huo, Z., Zhu, F. and Wang, Z. 2020. Fe-activated peroxymonosulfate enhances the degradation of dibutyl phthalate on ground quartz sand. *Environmental Science & Technology* 54(14), 9052-9061.
- Zhou, X., Lan, J., Liu, G., Deng, K., Yang, Y., Nie, G., Yu, J. and Zhi, L. 2012. Facet-mediated photodegradation of organic dye over hematite architectures by visible light. *Angewandte Chemie International Edition* 51(1), 178-182.
- Zong, M., Song, D., Zhang, X., Huang, X., Lu, X. and Rosso, K.M. 2021. Facet-dependent

photodegradation of methylene blue by hematite nanoplates in visible light. *Environmental Science & Technology* 55(1), 677-688.

Reviewer #2 (Remarks to the Author):

Wu et al. investigated the facet-dependent hydrolysis of phthalates over hematite nanoparticles. The authors prepared three well-defined nanoparticles with predominantly exposed, {104}, and {012} facets using hydrothermal methods. They demonstrated that the catalytic performance of hematite showed a significant facet-governing reactivity in the order {012} > {104} >> {001}. They used different methods to explain the facet-dependent mechanism, such as IR and cluster theoretical modeling. They concluded that proper neighboring Fe-Fe distance of 0.37 nm favored the formation of bidentate-coordinate with PAEs and thus promoted PAEs hydrolysis. The facet-dependent reactivity is different from the traditional hydrolysis mechanism, providing new insights into the structure-activity relationships. The manuscript is well organized, and the results support the conclusion, although there are some issues. Overall, the manuscript meets the scope of the journal. In my opinion, the manuscript can be published if the authors can make a thorough revision. The specific questions and concerns are summarized below.

Major

1. How did the authors confirm the atomic arrangements of these exposed surfaces? Many papers discuss the surface atomic arrangements for hematite, which is complicated. For instance, the hematite {001} has three atomic terminations: O termination, one-layer Fe termination, and double-layer Fe termination. The density of the undercoordinated Fe atoms should be different for different atomic terminations. However, such important information was not included and discussed in the manuscript.

Response: Thank you for the comments. The surface terminations of hematite {001} and {102} facets have been intensively investigated by either well-defined experiments or theoretical calculations (Cao et al., 2019; Kraushofer et al., 2018; Nguyen et al., 2013; Ovcharenko et al., 2016; Yan et al., 2020). In our original manuscript, we directly used the generally accepted surface termination model for the individual facet according to

these references. For example, for hematite {001} facet, the one-layer Fe-termination possesses the lowest surface energy under a wide range of oxygen chemical potentials (Figure R2, Layer 1), while the double-layer Fe termination (Figure R2, Layer 3) is usually energetically unfavorable (Nguyen et al., 2013; Ovcharenko et al., 2016; Schöttner et al., 2019; Yan et al., 2020). Other investigations by scanning tunneling microscopy (STM) proposed that both one-layer Fe-termination and O-termination might co-exist if the hematite {001} was prepared under high oxygen pressure (Bergermayer et al., 2004; Wang et al., 1998), or if the Fe atoms were to dissolve from an Fe-termination (Eggleston et al., 2003). Additionally, a ferryl (Fe=O) termination could also be observed under intermediate oxygen pressure, with the domain of one-layer Fe-termination (Lemire et al., 2005). While, the ferryl (Fe=O) termination exhibits higher surface energy than the one-layer Fe termination (Ovcharenko et al., 2016). Although the exact surface termination of {001} facet is under debate among different experimental and theoretical investigations, the one-layer Fe-termination is more preferential, due to its lower surface energy and non-polar characteristic after its surface relaxation (Schöttner et al., 2019; Voloshina, 2018). For {012} facet, its surface termination is relatively simple. It possesses one plausible stoichiometric (1×1) surface termination with the arm-chair like topography as depicted in Figure R3, Layer 1 (Cao et al., 2019; Kraushofer et al., 2018). In some conditions, the stoichiometric (1×1) surface can be reduced to (2×1) model by removing a row of bridging oxygen ions (Kraushofer et al., 2018), but it is reversible via O₂ treatment (Henderson, 2010). However, the surface termination of {104} facet is less studied (Chan et al., 2015; Zhang et al., 2021), and additional experiments and theoretical calculations may be required.

Therefore, in the revised manuscript, we calculated the surface energies of different possible surface terminations for each facet using DFT+*U* method, performed in the VASP software package version 5.4. The generalized gradient approximation (GGA) approach with the Perdew-Burke-Ernzerhof (PBE) exchange-correlation functional was used for the energy calculation. The Coulomb repulsion parameter

applied to the Fe $3d$ orbital of hematite was set as $U = 5.0$ eV. As shown in Table R1, the terminations of {001}-layer 1, {104}-layer 1/layer 5 and {012}-layer 1 possess the relatively lower surface energies.

In detail, for the {001} facet, its layer 1 termination is the most stable configuration that is in accordance with the previous reports (Nguyen et al., 2013; Ovcharenko et al., 2016; Schöttner et al., 2019; Yan et al., 2020). The outmost Fe atoms are relaxed inward to the underlayer plane of O atoms to reduce the surface energy (Figure R2, Layer 1). The inward relaxation is accompanied with significant charge redistribution, leading to the lowest empty surface state with Fe $3d_{z^2}$ characteristic, which also significantly reduces the surface dipole and increases its stability (Ovcharenko et al., 2016). The Bader charge analysis indicates that the outmost Fe atoms remain 6.39 valence electrons (VEs), which means that 1.61 (calculated as: $8-6.39$) VEs are transferred from Fe to the surrounding O anions (Figure R5a). Therefore, although the outmost Fe atoms are three-fold coordinated with O ligands, their Lewis-acidity is moderate.

For the {012} facet, the DFT+ U calculation suggests that the {012}-layer 1 is the most stable surface (Figure R3, Table R1), which is also in coincide with the previous studies (Cao et al., 2019; Kraushofer et al., 2018). After surface relaxation, the neighboring Fe distance is adjusted to 3.532 and 3.919 Å, which is suitable to form bidentate coordination with PAEs (Figure R6a). The surface Fe on {012}-layer 1 has the average VEs of 6.28, suggesting the relatively weak Lewis-acidity. However, the diffused reflectance infrared Fourier transform spectroscopy (DRIFTS) measurement shows that the {012} facet has even stronger Lewis acidity than the {001} and {104} facets (Please see the response to the reviewer 3, Figure 26 or Supplementary Figure 15). This discrepancy might be ascribed to the existence of surface O vacancies. Theoretically, the stoichiometric (1×1) surface of {012} facet is an ideal bulk termination. A certain degree of O defect must exist in actual sample (Kraushofer et al., 2018). We calculated the {012}-layer 1 termination with one O vacancy. The Fe atoms jointed to the O vacancy have the high VEs of 6.70 (Figure R5b), indicating much stronger Lewis-acidity. The surrounding Fe atoms also possess higher VEs, due to

surface charge re-distribution (Figure R5b).

For the {104} facet, we calculated the five possible terminations (Figure R4). The {104}-layer 1 possesses the relatively lower surface energy ($\gamma = 1.94 \text{ J/m}^2$), following with {104}-layer 5 ($\gamma = 2.12 \text{ J/m}^2$, Table R1). In comparison with the {104}-layer 1, the layer 5 termination has an additional layer of bridging O atoms between the outmost Fe atoms and the subsurface Fe atoms. From our understanding, the {104}-layer 5 could be regarded as the {104}-layer 1 after reacting with dissociated water molecules. Thus, the {104}-layer 1 and layer 5 may co-exist as the {104} facet termination, with the domain of layer 1 termination. After surface relaxation, on the {104}-layer 1, the outmost Fe and the subsurface Fe possess the VEs of 6.36 and 6.58 in average, respectively (Figure R5c). Since the subsurface Fe possesses stronger Lewis-acidity, it can be more attractive for ligands. Therefore, we further calculated the adsorption configuration of DMP on the {104}-layer 1 slab model. The initial geometry guess anchors the DMP molecule on the top of the slab, with two carbonyl groups orientating to the two outmost Fe atoms $\sim 2.0 \text{ \AA}$ above the slab. After optimization, one of the coordination sites always locates at the bridging position between one outmost Fe and one subsurface Fe, where the bridging O places on the {104}-layer 5 termination (Figure R6b). This result is essential for calculating the bidentate site density.

Moreover, the bare exposed Fe atoms are reactive with water molecules. Therefore, the actual hematite surface must be hydroxylated and hydrated. Many theoretical calculations and experimental studies have investigated the interaction between water molecules and the hematite {001} and {012} facets (Kraushofer et al., 2018; Nguyen et al., 2013; Ovcharenko et al., 2016; Schöttner et al., 2019; Tanwar et al., 2007; Voloshina, 2018). However, the surface hydroxylation and hydration conditions are complicated, and the reported results sometimes disagree with each other (Nguyen et al., 2013; Ovcharenko et al., 2016; Tanwar et al., 2007). By analyzing the O 1s XPS, the three hematite surfaces are partly hydroxylated and hydrated, with the bond -OH of 16-21.3%, and the chemisorbed H₂O of 13.0 - 17.3%, even under vacuum condition. Therefore, under ambient condition, the hematite surface should be more hydroxylated

and hydrated. However, PAEs coordinate with the surface exposed Fe rather than -OH/H₂O. If the Fe sites are occupied by hydroxyl groups or H₂O molecules, PAE molecules could compete/substitute with the surface bond -OH/H₂O for the uncoordinated Fe, especially under low surface moisture conditions. This can be evidenced by *in-situ* DRIFTS measurements that during Cl-DMA or DMP adsorbing onto the hematite surface, the surface bonded -OH/H₂O is simultaneously substituted as indicated by the reversal peaks in the range of 3000-3700 cm⁻¹ (please see Supplementary Figure 15 & 16). We believe that surface hydroxylation and hydration would not significantly change the coordination mode. Therefore, to simply the facet model, surface hydroxylation and hydration are not included when calculating the adsorption configuration and theoretical IR spectra.

In the revised manuscript, the corresponding “results and discussion” about the surface termination is provided in page 10-13, line 172-235: **“Determination of facet terminations.** The catalytic activity of hematite typically depends on its surface properties. However, the {001}, {104}, {012} facets have multiple theoretical surface terminations (Supplementary Fig. 11 - Fig. 13). To determine the exact termination, we calculated the surface energies of different possible terminations for each facet. It is generally accepted that the facet with the lowest surface energy is thermodynamically favorable. For the {001} facet, after surface relaxation, its layer 1 termination (Fe₃-O₃-Fe₆-R, the subscript represents for the coordination number, Fig. 3a) possesses much lower surface energy ($\gamma = 1.86 \text{ J}\cdot\text{m}^{-2}$) than the other terminations (Supplementary Table 1), which is in accordance with the previous reports⁴⁹⁻⁵¹. The exposed Fe atoms on {001}-layer 1 termination are three-fold coordinated. However, according to Bader charge analysis, the valence electrons (VEs) remaining on the exposed Fe are relatively low (VEs = 6.39, indicating 1.61 VEs are transferred from Fe to the adjacent O) (Fig. 3a), since the exposed Fe atoms are relaxed inward to the underlayer plane of O atoms with significant charge redistribution.

For the {012} facet, the stoichiometric termination with arm-chair like topology (O₃-Fe₅-O₄-Fe₆-O₄-R, Fig. 3c) was calculated as the most stable surface ($\gamma = 1.95 \text{ J}\cdot\text{m}^{-2}$

², Supplementary Table 1), which is consistent with the prior studies^{52,53}. Since its exposed Fe atoms are five-fold coordinated, the VE number is expectedly low (VE = 6.29, Supplementary Fig. 14d).

Compared to the other two facets, {104} facet was less investigated^{54,55}. In this study, the {104}-layer 1 termination (O₃-Fe₄-Fe₄-O₄-O₄-R, Fig. 3b) is calculated with the lowest surface energy ($\gamma = 1.94 \text{ J}\cdot\text{m}^{-2}$), following with the {104}-layer 5 termination (O₂-O₃-Fe₅-Fe₆-O₃-R, $\gamma = 2.12 \text{ J}\cdot\text{m}^{-2}$) (Supplementary Fig. 12). As both terminations possess the relatively low γ , they may co-exist with the domain of {104}-layer 1 as the stable {104} termination. The calculated VEs for the subsurface Fe (6.58) are higher than the outmost Fe (6.36) on the {104}-layer 1 (Fig. 3b).

It is important to note that the higher VEs of Fe cations usually implies lower coordination state and stronger coordination ability. However, when we further applied the diffused reflectance infrared Fourier transform spectroscopy (DRIFTS), using the compound 2-chloro-N,N-dimethylacetamide (Cl-DMA) as the probe molecule, to identify the surface Lewis-acid sites¹², the results are discrepant to the Bader charge analysis (Fig. 3, Supplementary Fig. 14). For example, HNC exhibits wider red-shift of $\nu_{\text{C=O}}$, which denotes to the carbonyl stretching vibration (1621 cm^{-1} on HNC vs. 1628 cm^{-1} on HNP & 1625 cm^{-1} on HNR, Supplementary Fig. 15a-c), suggesting that the HNC surface is more Lewis-acidic. This discrepancy might be ascribed to the existence of surface O-defects on actual HNC surface⁵³. By introducing a certain degree of O-defects on the surface of {012} facet model, the Fe atoms adjacent to the O-vacancies possess 6.70 - 6.74 VEs (Fig. 3c, Supplementary Fig. 14d), corresponding to the stronger Lewis-acidity. Meanwhile, a better fit for the IR spectra was also obtained using the {012} facet with O-defect for theoretical modelling (more discuss is shown below). Further study shows that the HNC400 (i.e., HNC calcinated at 400 °C for 2 h) exhibits a new shoulder peak at 1582 cm^{-1} (Supplementary Fig. 15f), probably stemming from the thermal desorption of -OH groups and H₂O molecules at the defective sites⁵⁶, suggesting a small amount of surface defective sites. For comparison, the $\nu_{\text{C=O}}$ peaks in both HNR and HNR400 samples are narrow and sharp

(Supplementary Fig. 15b, e), indicating negligible surface defects on HNR.

Moreover, the surface undercoordinated Fe sites are reactive with water molecules. Under low water partial pressure, H₂O molecules would entirely or dissociatively bond to lattice Fe and O with interactive hydrogen bonding, leading to surface hydroxylation, hydration and protonation (Fig. 3, Supplementary Fig. 14)⁵⁷⁻⁶⁰. The O 1s signals in XPS spectra demonstrate that the HNP, HNR and HNC surfaces are partially hydroxylated and hydrated (bond -OH: 16-21.3%, chemisorbed H₂O: 13.0 - 17.3%) even under vacuum condition (Supplementary Fig. 4 - Fig. 6). Therefore, under ambient condition, their surface should be more hydroxylated and hydrated⁶¹. Although surface hydroxylation/hydration could shield the surface reaction, and affect the valency of the exposed Fe, organic ligands can still compete with the surface bond -OH/H₂O for the undercoordinated Fe sites, especially under low moisture condition⁶², which is strongly evidenced by the desorption of the chemisorbed -OH/H₂O (3000 - 3750 cm⁻¹) substituted by either Cl-DMA or DMP on the three facets (Supplementary Fig. 15 & 16). Therefore, the surface hydroxylation/hydration would not change the coordination mode and the catalytic mechanism. To simplify the facet model, surface hydroxylation/hydration was not involved for modelling. In current study, the {001}-layer 1, {104}-layer 1 co-existing with layer 5, and the {012}-layer 1 with O-defects are deduced as the surface terminations for HNP, HNR and HNC, respectively (Fig. 3).”

In the “Methods” part, the method for VASP calculation was added, in page 24, line 479-488: “DFT calculations were performed to obtain the surface energies, the adsorption configurations, and the corresponding theoretical IR spectra, using the VASP 5.4 software. The generalized gradient approximation (GGA) approach with the Perdew-Bruke-Ernzerhof (PBE) exchange-correlation functional, and a cutoff energy of 550 eV for the planewave basis set were used for all the calculations. DFT + *U* method was adopted to treat the 3d orbital electrons of Fe with $U_{\text{eff}} = 5.00$ eV³¹. Core electrons are represented by the Project Augmented Wave (PAW) pseudopotentials with 8 and 6 valance electrons for Fe and O, respectively. Vibration analysis was performed according to the method provided by David Karhánek⁶⁷, and the IR spectrum was

plotted by the Multiwfn software⁶⁸.

Figure R2. The possible surface terminations of the hematite {001} facet.

Figure R3. The possible surface terminations of the hematite {012} facet.

Figure R4. The possible surface terminations of the hematite {104} facet.

Figure R5. Bader charge analysis of the facet-exposed Fe. The valence electron (VE) numbers on the outmost Fe atoms are marked in light yellow, and those on the subsurface Fe atoms are marked in sky blue. The green disks indicate the O vacancies.

Figure R6. Different adsorption configurations of DMP on the {012} facet slab (a) and the {104} facet slab (b).

Table R1. Surface energies (γ) for individual surface termination of different hematite facets.

Facet termination	E_{bulk} (eV)	$E_{\text{slab}}^{\text{unrelax}}$ (eV)	$E_{\text{slab}}^{\text{relax}}$ (eV)	A (\AA^2)	γ^a (J/m ²)
{001}-Layer 1		-515.17	-518.90		1.86
{001}-Layer 2	-203.46	-497.34	-499.50	88.73	3.69
{001}-Layer 3		-495.91	-298.13		3.81
{014}-Layer 1		-767.80	-772.55		1.94
{014}-Layer 2		-727.21	-734.82		3.80
{014}-Layer 3	-203.46	-761.14	-763.57	150.60	2.54
{014}-Layer 4		-727.53	-732.18		4.10
{014}-Layer 5		771.51	772.24		2.12
{012}-Layer 1		-781.18	-784.08		1.95
{012}-Layer 2		-758.92	-759.20		3.94
{012}-Layer 3	-203.46	-768.99	-769.86	110.38	3.13
{012}-Layer 4		-770.49	-773.70		2.68
{012}-Layer 5		-751.03	-752.30		4.37

a. The surface energy (γ) comprised by the cleavage energy (E_{cut}) and the reconstruction energy (E_{relax}) (Huang et al., 2021). All the atoms are kept fixed except for the atoms in the top surface layer. Namely, $\gamma = E_{\text{cut}} + E_{\text{relax}} = \frac{1}{2A}(E_{\text{slab}}^{\text{unrelax}} - nE_{\text{bulk}}) + \frac{1}{A}(E_{\text{slab}}^{\text{relax}} - E_{\text{slab}}^{\text{unrelax}})$, where $E_{\text{slab}}^{\text{unrelax}}$ is the unrelaxed energy of the fixed slab; E_{bulk} is the energy of the fully optimized bulk structure ($\text{Fe}_{12}\text{O}_{18}$); $2A$ denotes to the area of top and bottom surfaces after the cleavage; n is the total number of $\text{Fe}_{12}\text{O}_{18}$ units contained in the slab model; $E_{\text{slab}}^{\text{relax}}$ is the relaxed energy of the optimized slab with only the top surface relaxed.

References:

- Bergermayer, W., Schweiger, H. and Wimmer, E. 2004. Ab initio thermodynamics of oxide surfaces: O₂ on Fe₂O₃(0001). *Physical Review B* 69(19), 195409.
- Cao, S., Zhang, X., Huang, X., Wan, S., An, X., Jia, F. and Zhang, L. 2019. Insights into the facet-dependent adsorption of phenylarsonic acid on hematite nanocrystals. *Environmental Science: Nano* 6(11), 3280-3291.
- Chan, J.Y.T., Ang, S.Y., Ye, E.Y., Sullivan, M., Zhang, J. and Lin, M. 2015. Heterogeneous photo-Fenton reaction on hematite (α -Fe₂O₃){104}, {113} and {001} surface facets. *Physical Chemistry Chemical Physics* 17(38), 25333-25341.
- Eggleston, C.M., Stack, A.G., Rosso, K.M., Higgins, S.R., Bice, A.M., Boese, S.W., Pribyl, R.D. and Nichols, J.J. 2003. The structure of hematite (α -Fe₂O₃) (001) surfaces in aqueous media: scanning tunneling microscopy and resonant tunneling calculations of coexisting O and Fe terminations. *Geochimica et Cosmochimica Acta* 67(5), 985-1000.
- Henderson, M.A. 2010. Low temperature oxidation of Fe²⁺ surface sites on the (2×1) reconstructed surface of α -Fe₂O₃(011-2). *Surface Science* 604(13), 1197-1201.
- Huang, M., Liu, C., Cui, P., Wu, T., Feng, X., Huang, H., Zhou, J. and Wang, Y. 2021. Facet-dependent photoinduced transformation of cadmium sulfide (CdS) nanoparticles. *Environmental Science & Technology* 55(19), 13132-13141.
- Kraushofer, F., Jakub, Z., Bichler, M., Hulva, J., Drmota, P., Weinold, M., Schmid, M., Setvin, M., Diebold, U., Blaha, P. and Parkinson, G.S. 2018. Atomic-scale structure of the hematite α -Fe₂O₃(1 $\bar{1}$ 02) “R-Cut” surface. *The Journal of Physical Chemistry C* 122(3), 1657-1669.
- Lemire, C., Bertarione, S., Zecchina, A., Scarano, D., Chaka, A., Shaikhutdinov, S. and Freund, H.J. 2005. Ferryl (Fe=O) termination of the hematite α -Fe₂O₃(0001) surface. *Physical Review Letters* 94(16), 166101.
- Nguyen, M.T., Seriani, N. and Gebauer, R. 2013. Water adsorption and dissociation on α -Fe₂O₃(0001): PBE+U calculations. *The Journal of Chemical Physics* 138(19), 194709.
- Ovcharenko, R., Voloshina, E. and Sauer, J. 2016. Water adsorption and O-defect formation on Fe₂O₃(0001) surfaces. *Physical Chemistry Chemical Physics* 18(36), 25560-25568.
- Schöttner, L., Ovcharenko, R., Nefedov, A., Voloshina, E., Wang, Y., Sauer, J. and Wöll, C. 2019. Interaction of water molecules with the α -Fe₂O₃(0001) surface: A combined experimental and computational study. *The Journal of Physical Chemistry C* 123(13), 8324-8335.
- Tanwar, K.S., Lo, C.S., Eng, P.J., Catalano, J.G., Walko, D.A., Brown, G.E., Waychunas, G.A., Chaka, A.M. and Trainor, T.P. 2007. Surface diffraction study of the hydrated hematite (1102) surface. *Surface Science* 601(2), 460-474.
- Voloshina, E. (2018) *Encyclopedia of Interfacial Chemistry*. Wandelt, K. (ed), pp. 115-121, Elsevier, Oxford.
- Wang, X.G., Weiss, W., Shaikhutdinov, S.K., Ritter, M., Petersen, M., Wagner, F., Schlögl, R. and Scheffler, M. 1998. The hematite (α -Fe₂O₃)(0001) surface: Evidence for domains of distinct chemistry. *Physical Review Letters* 81(5), 1038-1041.
- Yan, L., Chan, T. and Jing, C. 2020. Arsenic adsorption on hematite facets: spectroscopy and DFT study. *Environmental Science: Nano* 7(12), 3927-3939.
- Zhang, H., Xu, Z., Chen, D., Hu, B., Zhou, Q., Chen, S., Li, S., Sun, W. and Zhang, C. 2021.

Adsorption mechanism of water molecules on hematite (104) surface and the hydration microstructure. *Applied Surface Science* 550, 149328.

2. As the reactivity is correlated to the surface with active sites, the reviewer would suggest that the turnover frequency (TOF) of the catalysts based on each Fe active site and turn number (TON) of the contaminants need to be compared for the three surfaces.

Response: Thank you for the comments. The turnover number (TON) is defined as the total mole of substrate converted by per mole of active site during a given reaction period. The turnover frequency (TOF) is the TON per reaction time, which could also be calculated from the reaction rate (zero order) per active site. Either TON or TOF could indicate the reactivity of the catalyst. In this study, the calculated TOF and TON are provided in Table R2. Similar TON and TOF values were obtained for DMP/DnBP reacting on HNR and HNC, suggesting the similar mechanism and comparable catalytic reactivity of HNR and HNC. While, the TON and TOF values for DMP/DnBP on HNP are one-order of magnitude lower compared to those on HNR and HNC, indicating a different surface interaction. According to our experimental results, the hydrolysis of DMP/DnBP on HNR and HNC is dominated by bidentate coordination, while, monodentate coordination is responsible for the reaction of DMP/DnBP on HNP. Moreover, for MB, since only monodentate coordination mode is formed on all the three facets, similar TON and TOF values were obtained on the three facets. The TOF/TON results suggest the similar Lewis acidity of the single Fe site on the three facets, while, bidentate coordination would contribute much stronger Lewis-acid catalytic activity than monodentate coordination.

In the revised manuscript, the relevant results and discussion have been added in page 14-15, line 272-284: “The catalytic activity of the facet also strongly depends on the affinity of the exposed Fe atoms. The *in-situ* DRIFTS measurement (Supplementary Fig. 15) and the Bader charge analysis (Fig. 3, Supplementary Fig. 14) have implied the surface Lewis-acid properties of the three facets. Herein, we further calculated the turnover numbers (TONs) and turnover frequencies (TOFs) of MB, DMP and DnBP on

the three facets, which could reflect the catalytic activity of single Fe site, regardless of the site density³¹. As listed in Supplementary Table 3, the reaction of MB on the three facets shows almost equal TOF/TON values, suggesting the similar Lewis acidity of the single Fe site on the three facets. However, for the reaction of DMP/DnBP, the obtained TON/TOF values are identical on HNR and HNC, but are one-order of magnitude higher than that on HNP. This result also provides strong evidence for our hypothesis that the hydrolysis of PAEs on HNR and HNC is ruled by bidentate coordination mechanism, which could induce much stronger Lewis-acid catalysis.”

Table R2. The calculated TON and TOF values of the hematite (HNP, HNR, HNC) for the degradation of DMP, DnBP and MB under RH 76%.

	k_{BET} ($10^{-2} \text{ g} \cdot \text{day}^{-1} \cdot \text{m}^{-2}$)			Surface area ($\text{m}^2 \cdot \text{g}^{-1}$)	Single Site density ($\text{site} \cdot \text{m}^{-2}$)	Couple site density ($\text{site} \cdot \text{m}^{-2}$)	TON (10^{-3}) ^a			TOF (10^{-3} day^{-1}) ^b		
	DMP	DnBP	MB				DMP	DnBP	MB	DMP	DnBP	MB
HNP	0.191	0.051	7.904	24.9	4.51	NA	0.50 ^c	0.14 ^c	9.21 ^c	0.51 ^c	0.14 ^c	20.9 ^c
HNR	2.038	1.170	16.506	11.4	10.62	7.97	2.75 ^d	1.65 ^d	8.42 ^c	3.08 ^d	1.77 ^d	18.7 ^c
HNC	4.176	2.551	11.555	18.1	7.25	14.50	2.43 ^d	1.70 ^d	8.04 ^c	3.47 ^d	2.12 ^d	19.1 ^c

Note: *a.* $\text{TON} = \frac{C_0 \cdot (1 - \exp(-k_{BET} \cdot \text{SSA} \cdot t)) \cdot N_A}{D_{si} \text{ or } D_{p-di} \cdot \text{SSA}}$; The TON was calculated after reacting for 1 d, $t = 1$ d; C_0 is the initial concentration of $2 \mu\text{mol} \cdot \text{g}^{-1}$; N_A

$= 6.02 \times 10^{23}$.

b. $\text{TOF} = \frac{C_0 \cdot k_{BET} \cdot N_A}{D_{si} \text{ or } D_{p-di}}$; The TOF of the initial stage is calculated.

c. These TOF and TON values are calculated based on the D_{si} .

d. These TOF and TON values are calculated based on the D_{p-bi} .

3. The authors used some organics (or additives) to prepare the hematite nanoparticles. Whether the organic ligands are completely removed needs to be confirmed. The contaminated organic carbon might influence the interfacial structures of phthalates complexed on these surfaces due to the possible competitive adsorption. The C 1s XPS should be feasible to prove whether the hematite surface was contaminated with carbon. What is more, how to wash the samples needs to be provided.

Response: Thank you for the comments. The X-ray photoelectron spectra (XPS) of the three hematites (HNP, HNR and HNC) are shown in Figure R7-R9 and all present clear C 1s peak. However, it does not mean that the hematite surface is heavily contaminated by the organic ligands during synthesis, since the fitted C compositions (C-C/C-H: 73-77%, C-N/C-O: 12-18%, C=O: 9-11%) do not match the chemical compositions of the organic ligands (CH_3COO^- for HNP, H_2NCONH_2 and HCONH_2 for HNR, $\text{CH}_3\text{COONH}_4$ for HNC). For example, H_2NCONH_2 and HCONH_2 were used to synthesize HNR. While, there are no C-C and C-H bonds in the two ligands, and the synthesizing temperature of 160-180 °C is unlikely to cause carbonization of the organic ligands. For HNR and HNC, the organic ligands used for synthesis contain high amount of N element. Therefore, the detection of N 1s by XPS could also reflect whether the hematite surface was contaminated by the organic ligands or not. As shown in Figure R8 and R9, the N 1s XPS signal was not observed on HNR and HNC, suggesting that the hematite surfaces were not contaminated by the organic ligands from synthesis.

To further verify this, we washed the hematite nanoparticles with ethanol and water for more than 10 times, then calcinated them at 400 °C for 2 h. However, the additional washing and calcination treatments still cannot remove the C 1s signal. Referring to the literatures from different researchers, the residual organic carbon is commonly detectable in XPS analysis. For example, the synthesized hematite {001}, {012}, {110} nanoparticles also presented clear C 1s XPS signal in the literatures (Figure R10), demonstrating that carbon contamination is difficult to avoid (Guo et al.,

2021; Huang et al., 2016; Shen et al., 2020). Since the C compositions on the three hematite are very similar, the carbon contamination might stem from extra sources after synthesis. However, the discussion for the exact source of the carbon contamination is beyond the scope of current study, further investigation should be conducted.

As suggested by the other reviewer, we also measured the isoelectric point of the hematite nanoparticles, as the isoelectric point of hematite could be sensitive to the adsorbed ions and ligands. The isoelectric points for HNC, HNP and HNR were measured as 7.0 ~ 8.5, close to the values reported in the literatures. For example, the isoelectric point of hematite {001} was reported as ~7.8 (Li et al., 2020), and the measured equivalent point of zero potential for hematite {001} and {012} were in the range of 8.35~8.65 (Chatman et al., 2013). If the hematite surface is contaminated by organic carboxylic acid, the surface electrostatic potential would be significantly reduced. It was reported that the isoelectric point of hematite {001} would decrease to $\text{pH} < 3$ in the presence of 10 μM oleic acid (Quast, 2016). It is obviously not the case for HNP in our study, as we used acetic acid for synthesis. Thus, our HNP {001} is unlikely contaminated by acetic acid.

Based on the above discussion, the collective XPS and isoelectric points results show that the three synthesized hematite are pure, with slight adventitious carbon contamination. However, the composition of C=O species is low (9-11%). Since only the C=O species can compete for the Lewis-acid sites, the C-C/C-H and C-N/C-O species have poor coordination capability, the slight carbon contamination should have little influence on the hydrolysis reaction. It would not challenge the main conclusion of current study.

Actually, in our experiments, after synthesis, the three hematites were carefully washed by ethanol (> 98%) and pure water (18.2 M Ω) for at least 5 times for each washing step. After washing, the hematite suspension was centrifuged and freeze-dried. To remove surface adsorbed free water molecules, the hematite particles were then oven-dried at 105 °C for overnight before the kinetic experiment. The information for how to wash the sample is provided in the Supplementary Material, Text 4: “After

cooling down to room temperature, the synthesized HNP, HNR and HNC powders were washed by water and ethanol alternately for at least 5 times to remove residual ligands or ions. Then, the hematite nanoparticles were freeze-dried for further use.”

Moreover, the discussion about surface characterization, based on the XPS and zeta potential analysis, has also been added to the revised manuscript, in page 8, line 129-139: “Their isoelectric points were measured as pH 7.0 - 8.5 (Supplementary Fig. 3), close to the reported values^{17,47}. No N 1s signals in X-ray photoelectron spectra (XPS) were detected on HNR and HNC (Supplementary Fig. 5 & Fig. 6), suggesting that no organic ligands from synthesis remain on the hematite surfaces. Although the XPS shows clear C 1s signals, the C=O species, which is able to compete for the Lewis-acid sites, comprises only a low amount (9-11% of the total surface carbon content, Supplementary Fig. 4 - Fig. 6). Therefore, the isoelectric points and the XPS results suggest that the HNP, HNR, HNC surfaces are clean with small amount of adventitious carbon contamination, and the residual carbon is expected to have little influence on the surface reactions.”

More discussion regarding to the surface properties is provided in Supplementary Text 1: “**Characterization of the surface condition.** The isoelectric point of hematite can to some extent indicate its surface condition, because the isoelectric point of hematite could be sensitive to the adsorbed ions and ligands. Based on zeta (ζ) potential measurement (Supplementary Fig. 3), the isoelectric points for HNC, HNP and HNR nanoparticles were measured as 7.0 - 8.5, close to the values reported in the literatures. For example, the isoelectric point of hematite {001} was reported as $\sim 7.8^1$, and the measured equivalent point of zero potential for hematite {001} and {012} were in the range of 8.35~8.65². If the hematite surface was contaminated by organic carboxylic acid, the surface electrostatic potential would be significantly reduced. It was reported that the isoelectric point of hematite {001} would reduce to $\text{pH} < 3$ in the presence of 10 μM oleic acid³. It is obviously not the case for HNP in our study, as we used acetic acid for synthesis. Thus, the isoelectric points suggest that our synthesized hematite nanoparticles were unlikely contaminated by the organic ligands used in synthesis.

X-ray photoelectron spectroscopy (XPS) can provide extra evidence to examine whether the hematite surface was contaminated by organic. The XPS results of the three hematite (HNP, HNR and HNC) are shown in Supplementary Fig. 4-Fig. 6. Although all presented clear C 1s peak, the fitted C compositions (C-C/C-H: 73-77%, C-N/C-O: 12-18%, C=O: 9-11%) do not match the chemical compositions of the organic ligands (CH₃COO⁻ for HNP, H₂NCONH₂ and HCONH₂ for HNR, CH₃COONH₄ for HNC). For example, H₂NCONH₂ and HCONH₂ were used to synthesize HNR. While, there are no C-C and C-H bonds in the two ligands, and the synthesizing temperature of 160-180 °C is unlikely to cause carbonization of the organic ligands. In addition, since the organic ligands used for synthesis contain high amount of N element in the cases of HNR and HNC, the detection of N 1s by XPS could also reflect whether the hematite surface was contaminated by the organic ligands or not. As shown in Supplementary Fig. 5 & Fig. 6, no N 1s XPS signal was observed on HNR and HNC. Therefore, we can conclude the hematite surfaces are not contaminated by the organic ligands from synthesis.

To further verify this, we washed the hematite nanoparticles with ethanol and water for more than 10 times, then calcinated them at 400 °C for 2 h. However, the additional washing and calcination treatments still cannot remove the C 1s signal. Referring to the literatures from different researchers, the residual organic carbon is commonly detectable in XPS analysis. For example, the synthesized hematite {001}, {012}, {110} nanoparticles also presented clear C 1s XPS signal in the literatures⁴⁻⁶, demonstrating carbon contamination on hematite surface is difficult to avoid. Since the C compositions on the three hematite are very similar, the carbon contamination might stem from extra sources after synthesis. However, the discussion for the extra source of the carbon contamination is beyond the scope of this study, further investigation should be conducted.

Based on the above discussion, the collective XPS and isoelectric points results show that the three synthesized hematite are pure, with slight adventitious carbon contamination. However, the composition of C=O species is low (9-11%). Since only the C=O species can compete for the Lewis-acid sites, the C-C/C-H and C-N/C-O

species have poor coordination capability, the slight carbon contamination should have little influence on the hydrolysis reaction. It would not challenge the main conclusion of current study.”

Figure R7. The XPS measurements of HNP, including the survey spectra, high resolution spectra of Fe 2p, O 1s, and C 1s. The contributions of different oxygen species for simulating O 1s were fitted according to Schöttner et al. (2019). The composition of C species for C 1s refers to the NIST X-ray photoelectron spectroscopy database (<https://srdata.nist.gov/xps/Default.aspx>).

Figure R8. The XPS measurements of HNR, including the survey spectra, high resolution spectra of Fe 2p, O 1s, C 1s, and N 1s. The contributions of different oxygen species for simulating O 1s were fitted according to Schöttner et al. (2019). The composition of C species for C 1s refers to the NIST X-ray photoelectron spectroscopy database (<https://srdata.nist.gov/xps/Default.aspx>).

Figure R9. The XPS measurements of HNC, including the survey spectra, high resolution spectra of Fe 2p, O 1s, C 1s, and N 1s. The contributions of different oxygen species for simulating O 1s were fitted according to Schöttner et al. (2019). The composition of C species for C 1s refers to the NIST X-ray photoelectron spectroscopy database (<https://srdata.nist.gov/xps/Default.aspx>).

Figure R10. The XPS images of the hematite {001}, {012}, {100} cited from the literatures (Guo et al., 2021; Huang et al., 2016; Shen et al., 2020). The peak in the square box is the C 1s signal.

References:

- Chatman, S., Zarzycki, P. and Rosso, K.M. 2013. Surface potentials of (001), (012), (113) hematite (α - Fe_2O_3) crystal faces in aqueous solution. *Physical Chemistry Chemical Physics* 15(33), 13911-13921.
- Guo, T., Jiang, L., Wang, K., Li, Y., Huang, H., Wu, X. and Zhang, G. 2021. Efficient persulfate activation by hematite nanocrystals for degradation of organic pollutants under visible light irradiation: Facet-dependent catalytic performance and degradation mechanism. *Applied Catalysis B: Environmental* 286, 119883.
- Huang, X., Hou, X., Song, F., Zhao, J. and Zhang, L. 2016. Facet-dependent Cr(VI) adsorption of hematite nanocrystals. *Environmental Science & Technology* 50(4), 1964-1972.
- Li, T., Zhong, W., Jing, C., Li, X., Zhang, T., Jiang, C. and Chen, W. 2020. Enhanced hydrolysis of p-nitrophenyl phosphate by iron (hydr)oxide nanoparticles: Roles of exposed facets. *Environmental Science & Technology* 54(14), 8658-8667.
- Quast, K. 2016. The use of zeta potential to investigate the interaction of oleate on hematite. *Minerals Engineering* 85, 130-137.
- Shen, Z., Zhang, Z., Li, T., Yao, Q., Zhang, T. and Chen, W. 2020. Facet-dependent adsorption and fractionation of natural organic matter on crystalline metal oxide nanoparticles. *Environmental Science & Technology* 54(14), 8622-8631.
- Schöttner, L., Ovcharenko, R., Nefedov, A., Voloshina, E., Wang, Y., Sauer, J. and Wöll, C. 2019. Interaction of water molecules with the α - Fe_2O_3 (0001) surface: A combined experimental and computational study. *The Journal of Physical Chemistry C* 123(13), 8324-8335.

4. The authors used iron cluster models to simulate the iron-DMP/DnBP complexes; however, some problems exist. The actual surface of hematite is periodic, so clusters cannot represent the periodic surface. The experimental and theoretical IR did not match well with each other; the rational use of the cluster modeling needs to be clarified.

Response: Thank you for the comments. In the original manuscript, we calculated the IR spectra of the pure DMP/DnBP and the Fe chelated DMP/DnBP by Gaussian 09 software, which can only use the $\text{Fe}_n(\text{OH})_m$ cluster for modelling. Although $\text{Fe}_n(\text{OH})_m$ cluster cannot fully represent the hematite surface, the calculated IR spectra match well with our experimental results. The main conclusion is that bidentate coordination can induce wider blue shift of the carbonyl vibration, and promote more intensive catalytic hydrolysis. In the revised manuscript, we applied the periodic hematite slabs ($\text{Fe}_{72}\text{O}_{108}$ for {001}, $\text{Fe}_{72}\text{O}_{108}$ for {104}, $\text{Fe}_{108}\text{O}_{162}$ for {012}) with the vacuum space of 15 Å for modelling. The calculations were performed with the VASP 5.4., using DFT + U method. The generalized gradient approximation (GGA) approach and the Perdew-Burke-Ernzerhof (PBE) exchange-correlation functional were used for calculation. The Coulomb repulsion parameter applied to the Fe $3d$ orbital of hematite was set as $U_{\text{eff}} = 5.0$ eV (Yan et al., 2020). Vibration analysis was performed according to the method by David Karhánek (<https://github.com/dakarhanek/vasp-infrared-intensities>), and the IR spectrum was plotted by the Multiwfn software (Lu & Chen, 2012).

We firstly calculated the theoretical IR spectrum of pure DMP, as a benchmark. As shown in Figure R11, the IR spectrum of DMP by VASP calculation is in accordance with that by Gaussian calculation. Furthermore, by comparison with the experimental IR spectrum, both the VASP and Gaussian methods can fit well with the IR spectrum of DMP itself.

Then, we calculated the IR spectra of DMP adsorbed on the {001}, {104}, {012} slabs. The surface termination of each facet was determined by their lowest surface energies (Table R1). The surface hydroxylation and hydration were not involved to simplify the adsorption model. Therefore, the {001}-layer 1, the {104}-layer 1 & layer 5, the {012}-layer 1 with/without O-defects were subjected for simulation.

For DMP adsorbed on {001}-layer 1, both the *trans*-DMP and *cis*-DMP form mono-dentate coordination with the surface exposed Fe atom (Figure R9). The optimized *cis*-DMP geometry is not ready to form bidentate coordination on {001}-layer 1, since the distance to the neighboring Fe is 0.5063 nm that is too wide ([Figure](#)

R2). The calculated IR spectra show that the carbonyl stretching vibration (ν_2) red-shifts to 1640-1650 cm^{-1} (Figure R12), which is slightly weaker than the experimental result ($\nu_2 = 1621 \text{ cm}^{-1}$, Figure 4e₍₂₎ & 4f₍₁₎ in the manuscript). This can be probably explained by the presence of O defects on actual HNP surface. Unless prepared with extreme care, the synthesized hematite surface generally possesses a wide variety of defects. Consequently, the surface electronic structure would also be changed by point defects associated with surface nonstoichiometry (Voloshina, 2018). The presence of surface O vacancy would increase the valency of the neighboring Fe atom (Figure R15a), and induce stronger surface Lewis acidity.

For DMP adsorbed on the {104}-layer 1 and layer 5, both terminations can induce bidentate coordination with *cis*-DMP (Figure R13). Due to the steric hinderance, *cis*-DMP molecules prefer to bind with the outmost Fe on {104}-layer 1. However, with the stronger Lewis acidity (Figure R5c), the subsurface Fe atoms can provide more significant inducing effect on the electron-deficient carbonyl group. Therefore, the second binding site would locate at the position between Fe_{outmost} and Fe_{subsurface}. Correspondingly, the calculated carbonyl vibration greatly red-shifts to 1548 -1587 cm^{-1} (Figure R13), in quite accordance with the experimental IR results (Figure 4e₍₃₎ & 4f₍₂₎, in the manuscript). Since the {014}-layer 5 also has relatively low surface energy (Table R1), it may co-exist as {104} surface termination. However, the subsurface Fe atoms on {104}-layer 5 are fully coordinated, thus, DMP molecule can only bidentate-coordinate with the outmost Fe. In this model, the outmost Fe atoms possess relatively low VEs (6.28, Figure R15b). Correspondingly, the obtained carbonyl vibration shows a relatively small red-shift to 1627 cm^{-1} (Figure R13), compared to that on {104}-layer 1.

For DMP adsorbed on the {012}-layer 1, both the neighboring Fe sites with the distance of 0.36 nm and 0.39 nm can bidentate-coordinate with *cis*-DMP, and the corresponding carbonyl vibration red-shifts to 1607~1620 cm^{-1} (Figure R14). Likewise, the actual {012} facet surface also contains a certain degree of O vacancy. The Fe atoms adjacent to the O vacancy would show higher VEs than those in the bulk surface (6.70

vs. 6.28). Besides, due to the surface charge redistribution, the VEs of the Fe surrounding the O vacancy also increases (Figure R15c). Taking the {012}-layer 1 with one O vacancy for modelling, the obtained IR peak of ν_2 shifts to 1590 cm^{-1} (Figure R14), which can also fit with our experimental observation. We suppose DMP molecules at low surface concentration prefer to interact with the Fe sites around O vacancies.

To sum up, we additionally performed the VASP calculation to simulate the adsorption configurations and IR spectra of DMP on different slab models, which could represent the periodic surface. The calculations can fit well with the experimental results. In general, bidentate coordination can induce stronger Lewis-acid interaction than monodentate coordination, imposing wider red-shift of carbonyl vibration to lower than 1600 nm^{-1} . In addition, using the simplified Fe-hydroxyl clusters to simulate surface Lewis-acid interaction, carried out by Gaussian calculation, is also reliable.

In the revised manuscript, we added the results by VASP calculation and the corresponding discussion in page 15-18, line 297-353:

“To obtain further insights into the adsorption mechanism, we applied two methods to calculate the theoretical IR spectra, and compared with the experimental IR spectra.

Firstly, the slab models can represent for the periodic hematite surface. So, the obtained DMP-slab adsorption configurations are subjected for vibration analysis by Vienna ab initio simulation package (VASP) calculation (Fig. 4c). For *trans*-DMP/*cis*-DMP adsorbed on the {001}-layer 1, both show the $\nu_{\text{C=O}}$ at 1708/1700 (ν_1 , uncomplexed) and 1640/1650 cm^{-1} (ν_2 , complexed) (Fig. 4c(1), Supplementary Fig. 18a). For the {104} facet, *cis*-DMP adsorbed on the {104}-layer 1 shows much wider red-shift of the $\nu_{\text{C=O}}$, *i.e.*, at 1614 (ν_1 , complexed by the outmost Fe) and 1587 cm^{-1} (ν_2 , co-complexed by $\text{Fe}_{\text{subsurface}}$ and $\text{Fe}_{\text{outmost}}$) (Fig. 4c(2), Supplementary Fig. 18b-c). On the other hand, *cis*-DMP adsorbed on the {104}-layer 5 exhibits relatively weaker interaction with the $\nu_{\text{C=O}}$ at 1664 (ν_1)/1627 (ν_2) cm^{-1} (Supplementary Fig. 18d). For the {012} facet without O-defect, the $\nu_{\text{C=O}}$ of *cis*-DMP shows distinct absorption peaks at

1645 (ν_1) / 1625 (ν_2) cm^{-1} or 1674 (ν_1) / 1607 (ν_2) cm^{-1} (Supplementary Fig. 18e-f). While, on the O-defective site of {012}, the $\nu_{\text{C=O}}$ appears at 1627 (ν_1) and 1590 (ν_2) cm^{-1} (Fig. 4c(3)). Therefore, the theoretical IR spectra of DMP on the slab models directly show the coordination modes. In general, bidentate coordination can induce stronger Lewis-acid interaction than monodentate coordination, imposing wider red-shift of $\nu_{\text{C=O}}$ even to $< 1600 \text{ nm}^{-1}$.

Secondly, the simplified Fe-hydroxyl clusters were also introduced to complex with DMP/DnBP as the representatives of the surface coordination models (Fig. 4d), as the following hydrolysis pathways were calculated using the DMP-Fe-hydroxyl cluster models. This calculation was conducted by the Gaussian software. Taking DMP molecule itself as a benchmark, both the VASP and Gaussian calculations can precisely predict the IR spectrum of pure DMP (Supplementary Fig. 19). In detail, the symmetrical carbonyls of *trans*-DMP possess one theoretical $\nu_{\text{C=O}}$ at $\sim 1730 \text{ cm}^{-1}$ (Fig. 4d(1)). For *cis*-DMP, since its two carbonyls are asymmetric, the $\nu_{\text{C=O}}$ splits to 1724 and 1744 cm^{-1} (Fig. 4d(2)). When one carbonyl group of *trans*-DMP coordinates with $\text{Fe}(\text{OH})_3$, the un-complexed $\nu_{\text{C=O}}$ locates at 1700 (ν_1) cm^{-1} , and the complexed one red-shifts to 1645 (ν_2) cm^{-1} (Fig. 4d(3)). When both carbonyls of *cis*-DMP bidentate-coordinate with $\text{Fe}_2\text{O}(\text{OH})_4$, the $\nu_{\text{C=O}}$ groups exhibit much wider red-shift to 1645 (ν_1) and 1596 (ν_2) cm^{-1} (Fig. 4d(4)). Similar results were obtained for DnBP (Supplementary Fig. 20). The IR spectrum of DMP by Gaussian calculation is in accordance with that by VASP calculation. Therefore, using the simplified Fe-hydroxyl clusters to simulate surface Lewis-acid interaction is also reliable.

The experimental IR spectra of DMP/DnBP adsorbed on HNP, HNR, HNC were investigated by *ex-situ* IR and *in-situ* DRIFTS methods. For the *ex-situ* IR method, DMP/DnBP was initially applied onto the HNP/HNR/HNC nanoparticles, and the mixture was ground to prepared KBr wafers for measurement. As shown in Fig. 4e(2), the two $\nu_{\text{C=O}}$ peaks appear at 1724 cm^{-1} and 1626 cm^{-1} on HNP, with the red-shift of $\sim 98 \text{ cm}^{-1}$. Even wider red-shifts were observed on HNR ($\sim 137 \text{ cm}^{-1}$, Fig. 4e(3)) and HNC ($\sim 139 \text{ cm}^{-1}$, Fig. 4e(4)). Similar phenomenon was also observed for DnBP

(Supplementary Fig. 21).

By *in-situ* DRIFTS measurement, DMP was blown into the system filled with the HNP/HNR/HNC nanoparticles, then the cumulative adsorption of DMP on the hematite particles was *in-situ* detected. As shown in Fig. 4f, two significant $\nu_{C=O}$ absorption peaks were observed at ~ 1718 and 1620 cm^{-1} on all the three facets. The former indicates the physical adsorbed or weakly bond DMP, and the later should represent for the Lewis-acid coordinated one. After 130 min, two additional shoulder peaks at ~ 1565 and $\sim 1590\text{ cm}^{-1}$ start to appear on HNR (Fig. 4f(2), Supplementary Fig. 22a-b), suggesting bidentate-coordinated adsorption configuration formed on HNR. Such shoulder peaks were also observed on HNC (Fig. 4f(3)), however, did not appear on HNP (Fig. 4f(1)). Both the *ex-situ* and *in-situ* experimental IR spectra are in good accordance with the theoretical IR predictions (Fig. 4c & 4d), thus, providing the strong evidence that DMP is adsorbed onto HNR and HNC via the bidentate coordination, while via the monodentate coordination on HNP. As indicated by the much wider red-shifts of $\nu_{C=O}$, the bidentate coordination can induce stronger Lewis-acid interaction.”

Figure R11. The experimental IR spectrum of DMP (a), and the theoretical IR spectra of *trans*-DMP (b,d) and *cis*-DMP (c, e) by Gaussian calculation (b, c) and VASP calculation (d, e).

Figure R12. The adsorption models of *trans*-DMP (the first line) and *cis*-DMP (the second line) on the {001} facet, and the corresponding theoretical IR spectra.

Figure R13. The adsorption models of *cis*-DMP on the {104} facet layer 1 and layer 5, and the corresponding theoretical IR spectra.

Figure R14. The adsorption models of *cis*-DMP on the {012} facet layer 5 (the first line) and {104}-layer 1 (the second and third lines), and the corresponding theoretical IR spectra.

Figure R15. Bader charge analysis of the surface exposed Fe atoms for the {001} facet with one O vacancy (a), the {104} facet-layer 5 (b), the {012} facet-layer 1 with one O vacancy (c). VE means the valence electrons remaining on Fe.

References:

- Lu, T. & Chen, F. 2012. Multiwfn: A multifunctional wavefunction analyzer. *Journal of Computational Chemistry* 33, 580-592.
- Voloshina, E. (2018) *Encyclopedia of Interfacial Chemistry*. Wandelt, K. (ed), pp. 115-121, Elsevier, Oxford.
- Yan, L., Chan, T. & Jing, C. 2020. Arsenic adsorption on hematite facets: spectroscopy and DFT study. *Environmental Science: Nano* 7, 3927-3939.

Minor

1. *Are the configurations in figure 3a were checked by DFT calculation?*

Response: Thank you for the comment. In the revised manuscript, we calculated the adsorption configurations of DMP on the different slab models by DFT calculation in VASP, and the exact adsorption configurations are shown in **Figure 4a-b and Supplementary Figure 18**, as presented below.

Figure 4. Surface interaction modes of DMP on HNP/HNR/HNC. (a-b) The adsorption configurations of DMP on {001}-layer 1, {104}-layer 1, and {012}-layer 1 (with O-defect) slabs, respectively. (c) The corresponding theoretical IR spectra of DMP on the three facets. (d) The theoretical IR spectra of *trans*-/*cis*-DMP and DMP-Fe-hydroxyl complexes. (e) The experimental IR spectra of DMP before and after

adsorption onto HNP, HNR and HNC, using *ex-situ* KBr wafer method. **(d)** The *in-situ* DRIFTS of DMP adsorbed onto HNP, HNR and HNC under RH 76%.

Supplementary Figure 18. The adsorption configurations of DMP on {001}, {104} and {012} facets, as well as the corresponding calculated IR spectra.

2. The size distribution of these nanoparticles may be needed because sizes seem not well distributed from the SEM images, especially for the pseudo-cube nanoparticles.

Response: Thank you for the comment. The size distribution of synthesized hematite nanoparticles was characterized by the laser particle size analyzer (ZEN 3500 Zetasizer), at 100 mg/L and pH = 4.0 without pH buffer. The nanoparticles were dispersed directly into water and sonicated for 10 min before the measurement. As shown in Figure R16, the three hematites have wide size distribution. The average particle sizes for HNP, HNR and HNC are 304, 462 and 370 nm, slightly higher than the apparent particle sizes from SEM and TEM images, probably due to the partial aggregation of the nanoparticles in water.

In the revised manuscript, the corresponding information was added in page 8, line 128-129: “The particle size distributions of HNC, HNP, HNR are presented in Supplementary Fig. 2.”

Figure R16. The particle size distributions of HNP{001}, HNR{104} and HNC{012}.

3. Some related papers may be cited because they are conceptually related on a higher level, such as *Facet-dependent contaminant removal properties of hematite*

nanocrystals and their environmental implications. Environ. Sci.: Nano 2018, 5 (8), 1790-1806.

Response: Thank you for the comment. The paper “Facet-dependent contaminant removal properties of hematite nanocrystals and their environmental implications” has been cited in the revised manuscript as reference 26, in page 5, line 69-72: “As large quantities of engineered oxide mineral nanomaterials have been released into the environment^{24,25}, facet-dependent reactions would play unignorable roles in environmental/geochemical processes^{26,27}”

4. *The reactivity of the nanoparticles needs to be compared to other metal oxides for the hydrolysis of phthalates.*

Response: Thank you for the comment. In the revised manuscript, we applied another two common metal oxides, γ -Al₂O₃ and TiO₂ for comparison. The γ -Al₂O₃ was prepared from active neutral alumina (Shanghai Ludu Chemical Research Co. Ltd., Shanghai, China). Briefly, the neutral alumina was initially ground to pass through a 100 mesh sieve, then oven-heated at 600 °C for 2 h to obtain the γ -Al₂O₃. While, the TiO₂ nanopowder (CAS 13463-67-7), which is amorphous, was purchased from Shanghai Lingfeng Chemical Research Co. Ltd. (Shanghai, China). As shown in Figure R17, the γ -Al₂O₃ and the amorphous TiO₂ perform one-order of magnitude lower catalytic hydrolysis activity than HNR (0.232 day⁻¹) and HNC (0.755 day⁻¹), even slightly lower than HNP (0.0476 day⁻¹). We suppose that the over hydroxylated surface of γ -Al₂O₃ is unfavorable for Lewis-acid catalytic process. Moreover, since the TiO₂ nanopowder is amorphous, it is also unable to form the bidentate coordination with DMP.

As our study has indicated that the plane facet and the formation of bidentate coordination are the two important prerequisites for rapid hydrolysis performance, we then synthesized two TiO₂ nanoparticles with the main {001} facet and {101} facet,

respectively, according to the methods by Li, et al. (2015). From the SEM images (please see the response to Reviewer 1, Figure R1), the TiO₂-{001} and TiO₂-{101} nanoparticles were successfully synthesized. The neighboring Ti-Ti distances on both {001} and {101} facets are 0.38 nm, which is appropriate for bidentate coordination with DMP. Then, the TiO₂-{001} and TiO₂-{101} nanoparticles were subjected for degrading DMP under the same experimental conditions (RH 76%, 2.0 μmol·g⁻¹). As shown in Figure R18, both materials perform considerable (apparent) hydrolysis rates for DMP, even faster than HNR. From this point of view, our study can inspire researchers to develop more efficient metal oxide nanoparticles, which possess the proper surface atomic (i.e., the active sites) array. However, more investigations are still needed to solidify this mechanism.

In the revised manuscript, the result about γ-Al₂O₃ is not included. While, the results and discussion about amorphous TiO₂, TiO₂-{001} and TiO₂-{101} nanoparticles were added to page 21, line 418-431: “This mechanism would inspire researchers to develop novel nanomaterials with ideal surface atomic (i.e., the active sites) array to achieve higher catalytic performance. For example, we also synthesized the {101} and {001} facet-controlled TiO₂ nanoparticles for the same hydrolysis experiment (Supplementary Fig. 26a, d)⁶⁶. Meanwhile, one commercial amorphous TiO₂ nanopowder was used for comparison. The neighboring Ti-Ti distances on both {101} and {001} facets are 0.38 nm (Supplementary Fig. 26c, f), which is appropriate for bidentate coordination with DMP, according to our proposed mechanism. As shown in Supplementary Fig. 26g, both TiO₂-{101} and TiO₂-{001} perform considerable (apparent) hydrolysis reactivity for DMP, even more reactive than HNR. However, the catalytic activity of the amorphous TiO₂ nanopowder is much lower, suggesting that the plane facet and bidentate-coordination are the two important prerequisites for rapid surface-catalyzed hydrolysis reactions. While, more investigations are still needed to solidify this mechanism.”

Figure R17. The hydrolysis kinetics of DMP ($2.0 \mu\text{mol}\cdot\text{g}^{-1}$) on $\gamma\text{-Al}_2\text{O}_3$ and amorphous TiO_2 under RH 76%.

Figure R18. The hydrolysis kinetics of DMP ($2.0 \mu\text{mol}\cdot\text{g}^{-1}$) on $\text{TiO}_2\text{-}\{001\}$ and $\text{TiO}_2\text{-}\{101\}$.

{101} under RH 76%.

Reference:

Li, C., Koenigsmann, C., Ding, W., Rudshiteyn, B., Yang, K.R., Regan, K.P., Konezny, S.J., Batista, V.S., Brudvig, G.W., Schmittenmaer, C.A. and Kim, J.-H. 2015. Facet-dependent photoelectrochemical performance of TiO₂ nanostructures: An experimental and computational study. *Journal of the American Chemical Society* 137(4), 1520-1529.

5. *The iron atoms between two adjacent surfaces are also undercoordinated, and thus the contribution of the reactivity needs to be evaluated.*

Response: Thank you for the comment. The amount of edge Fe atoms between two adjacent facets should be far less than the amount of undercoordinated Fe atoms on the exposed facet, according to the relative surface areas (Figure R19) (Huang et al., 2019). Therefore, the possible contribution from the edge/corner sites should be insignificant.

The Fe atoms between two adjacent surfaces (namely, the Fe in edge sites) usually possess lower coordination number. So, they should have stronger Lewis-acidity, not only to the contaminants, but also to surface water molecules. According to Boily et al. (2015) and Liu et al. (1998), water molecules are reactive with the edge and defective sites under low water vapor pressure condition (Figure R20). When surface water molecule is more abundant than the contaminants, these undercoordinated sites would prefer to react with water molecules. Finally, the Lewis-acidity of the edge sites would be reduced to a homogenized level as the bulk surface sites. Therefore, the strong Lewis-acidity of the edge sites has the low probability to react with the contaminants, if the mineral surface contains more than one monolayer of water (at RH 15%).

Figure R19. The ideal shapes of HNP, HNR and HNC.

Figure R20. Simulation cell (a) and MD (b-e) snapshots (looking down the (001) facet) of OH-terminated $\sim 5 \times \sim 6 \times \sim 6$ nm α -Fe₂O₃ multi-facet particles. The sequence of snapshots at 1.5, 5.0 and 15 H₂O/nm² (b-d) illustrates the preferential accumulation of water molecules at the (100) facet, and the formation of a two-dimensional distribution of water molecules at the (001) facet. Water molecules located at the edges in the simulation of 5 H₂O/nm² are highlighted in panel e. This figure was cited from a reference (Boily et al., 2015).

References:

- Boily, J.-F., Yeşilbaş, M., Md. Musleh Uddin, M., Baiqing, L., Trushkina, Y. and Salazar-Alvarez, G. 2015. Thin water films at multifaceted hematite particle surfaces. *Langmuir* 31(48), 13127-13137.
- Huang, X., Chen, Y., Walter, E., Zong, M., Wang, Y., Zhang, X., Qafoku, O., Wang, Z. and Rosso, K.M. 2019. Facet-specific photocatalytic degradation of organics by heterogeneous fenton chemistry on hematite nanoparticles. *Environmental Science & Technology* 53(17), 10197-10207.
- Boily, J.-F., Yeşilbaş, M., Md. Musleh Uddin, M., Baiqing, L., Trushkina, Y. and Salazar-Alvarez, G. 2015. Thin Water Films at Multifaceted Hematite Particle Surfaces. *Langmuir* 31(48), 13127-13137.
- Liu, P., Kendelewicz, T., Brown, G.E., Nelson, E.J. and Chambers, S.A. 1998. Reaction of water vapor with α -Al₂O₃(0001) and α -Fe₂O₃(0001) surfaces: synchrotron X-ray photoemission studies and thermodynamic calculations. *Surface Science* 417(1), 53-65.

6. Line 117, dihedral angle with the adjacent lateral facet was 86°, not 90°. The nanocube nanoparticle is a pseudo-cube.

Response: Thank you for the comment. In the revised manuscript, page 8, line 125-

127, it has been modified: “For HNC, it has a pseudo-cubic shape with edge length of ~150 nm (Fig. 1c, f), lattice fringe of 0.37 nm and lattice angle of ~86° (Fig. 1i), indicating the single {012} facet.”

7. Figure 4, Please indicate what the atoms are in different colors.

Response: Thank you for the comment. In the revised manuscript, the atoms in different colors have been assigned in Figure 3 and Figure 4.

8. TOC art is not fully displayed.

Response: Thank you for the comment. In the revised manuscript, a new compact TOC is provided.

Figure R21. The new TOC art.

Reviewer #3 (Remarks to the Author):

The submitted work deals with the role of hematite crystals on the hydrolysis of phthalates in vapor phase, and especially the difference between the face orientations.

The title claims the “non-aqueous” hydrolysis, but the authors mean that the reaction take place in presence of hydrated vapor (relative humidity of 76%). So, the title should be changed into “(water) vapor phase” rather than “non-aqueous”.

Response: Thank you for the comments. It is indeed a tricky question how to define the concept of “non-aqueous reaction”. Originally, the description of “non-aqueous reaction” is to distinguish it from the “aqueous reaction”, because the majority of interfacial studies were conducted in water solution, without considering the actual moisture conditions even if necessary. The mineral surface under relatively dry conditions would exhibit significantly different surface properties in comparison with that in aqueous phase.

Under the exposure of ambient humidity, the mineral surface is not completely dry. It was reported that the mineral surface retains several layers of water molecules under the relative humidity (RH) of 76% (Boily et al., 2015; Wirth et al., 2016). It is difficult to describe such surface condition precisely and elegantly. We have used other expression, e.g., the “low-moisture condition”, “near drought/dry condition”, “water-unsaturated phase/condition”. In addition, the expression of “(water) vapor phase” is appropriate for describing the reactions in the atmosphere. However, we think that it may not be suitable to represent the soil condition. After a comprehensive consideration, in the revised manuscript, we modified the article title to “*The facet effect of hematite on the hydrolysis of phthalates under ambient humidity/moisture condition*”, and the expression of “non-aqueous” in the revised manuscript has also been modified.

References:

Boily, J.-F., Yeşilbaş, M., Md. Musleh Uddin, M., Baiqing, L., Trushkina, Y. and Salazar-Alvarez,

- G. 2015. Thin water films at multifaceted hematite particle surfaces. *Langmuir* 31(48), 13127-13137.
- Wirth, J., Kirsch, H., Wlosczyk, S., Tong, Y., Saalfrank, P. and Campen, R.K. 2016. Characterization of water dissociation on α -Al₂O₃(102): theory and experiment. *Physical Chemistry Chemical Physics* 18(22), 14822-14832.

The concept of face-dependent reactivity has existed for decades, as in the earlier work by Knozinger in catalysis (doi 10.1080/03602457808080878 for example in 1978), or Hiemstra in environmental sciences (doi 10.1016/0021-9797(89)90284-1 in 1989). Unfortunately, the bibliography of the submitted work only refer to the articles of 2000s (with very few exceptions). A few lines about the history of this approach would be welcomed.

Response: Thank you for the comments. Although the facet-dependent reactivity has been recognized for decades, the reports about the facet effect for catalysis or environmental application are scarce before 2010 (Figure R22). In the revised manuscript, a brief introduction about the earlier study on the facet effect for environmental application and nature processes has been added in page 5, line 66-79: “Although facet-dependent reactivities of oxide minerals have been recognized for decades¹⁸⁻²¹, the environmental implications (e.g., contamination removal) are getting more and more attentions in recent 10 years, due to the development and feasibility of surface science technologies^{22,23}. As large quantities of engineered oxide mineral nanomaterials have been released into the environment^{24,25}, facet-dependent reactions would play unignorable roles in environmental/geochemical processes^{26,27}. For hematite, its facet-dependent reactivity has been widely observed in multiple environmental fields, including iron dissolution²⁸, (in)organic substances adsorption^{29,30}, photocatalytic degradation³¹⁻³⁴, and thermo-catalytic oxidation³⁵, etc. However, almost all these facet-dependent reactions were found in aqueous phase^{17, 24-31}, and so far, only one study reported the facet-mediated hydrolysis reaction by hematite¹⁷. Therefore, exploring the hematite facet-mediated hydrolysis of PAEs under ambient humidity/moisture conditions can extend our understanding for the

environmental fate of PAEs.”

Figure R22. The number of year-published articles with the key words of “facet” + “catalysis”, or “facet” + “adsorption”. The data were collected from Web of Science (<https://www.webofscience.com/wos/alldb/basic-search>).

Detailed comments

Li. 106. The orientation of faces has been characterized by microscopy. The surface reactivity depends on the purity of these surfaces, so a low amount of adsorbed ions/molecules can change it. It would be interesting to measure the isoelectric point of this solids by zetametry, and compares them to calculated ones to ensure that the difference in reactivity comes from the orientation and not from the presence of surface residual chemicals used in the synthesis.

Response: Thank you for the comments. The three hematite after synthesis were carefully washed by ethanol (> 98%) and pure water (18 MΩ) for at least 5 times for each washing step. Usually, such washing protocol is adequate to remove the surface residual organic ligands and ions. To verify this, as suggested by the reviewer, we measured the surface zeta (ζ) potentials of the three hematite in a wide pH range (2-11). As shown in Figure R23, the measured isoelectric points for HNC, HNP and HNR are between pH 7.0 and 8.5, close to the values reported in the literatures (Chatman et al.,

2013; Li et al., 2020). For example, the isoelectric point of hematite {001} was reported as ~ 7.8 by the measurement of ζ potential (Li et al., 2020), and the measured equivalent point of zero potential for hematite {001} and {012} were in the range of 8.35~8.65 (Chatman et al., 2013). The isoelectric point of hematite can to some extent indicate its surface condition. If the hematite surface is contaminated by organic carboxylic acid, the surface electrostatic potential would be significantly reduced. For example, it was reported that the isoelectric point of hematite {001} would decrease to $\text{pH} < 3$ in the presence of $10 \mu\text{M}$ oleic acid (Quast, 2016). It is obviously not the case for HNP {001} in our study, as we used acetic acid for synthesis.

However, we did not find the theoretical isoelectric point of hematite from literatures. The $\text{p}K_{\text{a}}$ values of the surface hydroxyl groups on hematite {001} and {012} were once predicted to be 10.3~18.9 by Ab initio molecular dynamics modelling (Yan et al., 2020). While, it is difficult to calculate the isoelectric point from the theoretical $\text{p}K_{\text{a}}$ values, due to the surface heterogeneity.

In addition, we measured the C 1s and N 1s signals of the synthesized hematite using X-ray photoelectron spectroscopy (XPS), to explore whether the hematite surface was contaminated by organic molecules. The most likely source of carbon contamination is the residual organic ligands from synthesis. The organic ligands used for synthesizing HNR are H_2NCONH_2 and HCONH_2 , and $\text{CH}_3\text{COONH}_4$ for HNC. All the ligands contain high amount of N element. While, as shown in Figure R24a-b, no N 1s XPS signal was observed on HNR and HNC, suggesting that no organic ligands are left on the hematite surface after preparation. However, the C 1s signal shows slight carbon contamination (Figure R24c-e). The C 1s spectra on HNP, HNR and HNC are similar (C-C/C-H: 73-77%, C-N/C-O: 12-18%, C=O: 9-11%), and the compositions of carbon species do not match the chemical compositions of the organic ligands. For example, H_2NCONH_2 and HCONH_2 were used to synthesize HNR. While, there are no C-C and C-H bonds in H_2NCONH_2 and HCONH_2 , and the synthesizing temperature of 160-180 °C is unlikely to cause carbonization of the organic ligands. Therefore, such carbon contamination might be from some adventitious sources. Actually, carbon

contamination on hematite is difficult to avoid, which were easily detectable by XPS analysis, and were commonly reported in the literatures (Figure R10) (Guo et al., 2021; Huang et al., 2016; Shen et al., 2020). The discussion for the exact source of the carbon contamination is beyond the scope of current study.

However, it is important to note, the composition of C=O species only accounts for 9-11% of the total carbon component on the hematite surface. Only the C=O species can compete for the Lewis-acidic sites. The C-C/C-H and C-O/C-N species have poor complexation capability. Therefore, the slight carbon contamination should have little influence on the hydrolysis reaction. It would not challenge the main conclusion of current study. Therefore, the isoelectric points and the XPS results suggest the synthesized hematite surface are clean with small amount of adventitious carbon contamination. The varied surface reactivity is unlikely due to the presence of surface residual chemicals. Since MB and DMP/DnBP process different reaction kinetics on the three hematite, this provides a more robust justification that the difference in reactivity comes from the atomic-array of the facet.

The discussion about the surface condition has been added to the revised manuscript in page 8, line 129-139: “Their isoelectric points were measured as pH 7.0 - 8.5 (Supplementary Fig. 3), close to the reported values^{17,47}. No N 1s signals in X-ray photoelectron spectra (XPS) were detected on HNR and HNC (Supplementary Fig. 5 & Fig. 6), suggesting that no organic ligands from synthesis remain on the hematite surfaces. Although the XPS shows clear C 1s signals, the C=O species, which is able to compete for the Lewis-acid sites, comprise only a low amount (9-11% of the total surface carbon content, Supplementary Fig. 4 - Fig. 6). Therefore, the isoelectric points and the XPS results suggest that the HNP, HNR, HNC surfaces are clean with small amount of adventitious carbon contamination, and the residual carbon is expected to have little influence on the surface reactions.”

More discussion is provided in Supplementary Text 1: “**Characterization of the surface condition.** The isoelectric point of hematite can to some extent indicate its surface condition, because the isoelectric point of hematite could be sensitive to the

adsorbed ions and ligands. Based on zeta (ζ) potential measurement (Supplementary Fig. 3), the isoelectric points for HNC, HNP and HNR nanoparticles were measured as 7.0 - 8.5, close to the values reported in the literatures. For example, the isoelectric point of hematite {001} was reported as $\sim 7.8^1$, and the measured equivalent point of zero potential for hematite {001} and {012} were in the range of 8.35~8.65². If the hematite surface was contaminated by organic carboxylic acid, the surface electrostatic potential would be significantly reduced. It was reported that the isoelectric point of hematite {001} would reduce to $\text{pH} < 3$ in the presence of 10 μM oleic acid³. It is obviously not the case for HNP in our study, as we used acetic acid for synthesis. Thus, the isoelectric points suggest that our synthesized hematite nanoparticles were unlikely contaminated by the organic ligands used in synthesis.

X-ray photoelectron spectroscopy (XPS) can provide extra evidence to examine whether the hematite surface was contaminated by organic. The XPS results of the three hematite (HNP, HNR and HNC) are shown in Supplementary Fig. 4-Fig. 6. Although all presented clear C 1s peak, the fitted C compositions (C-C/C-H: 73-77%, C-N/C-O: 12-18%, C=O: 9-11%) do not match the chemical compositions of the organic ligands (CH_3COO^- for HNP, H_2NCONH_2 and HCONH_2 for HNR, $\text{CH}_3\text{COONH}_4$ for HNC). For example, H_2NCONH_2 and HCONH_2 were used to synthesize HNR. While, there are no C-C and C-H bonds in the two ligands, and the synthesizing temperature of 160-180 $^\circ\text{C}$ is unlikely to cause carbonization of the organic ligands. In addition, since the organic ligands used for synthesis contain high amount of N element in the cases of HNR and HNC, the detection of N 1s by XPS could also reflect whether the hematite surface was contaminated by the organic ligands or not. As shown in Supplementary Fig. 5 & Fig. 6, no N 1s XPS signal was observed on HNR and HNC. Therefore, we can conclude the hematite surfaces are not contaminated by the organic ligands from synthesis.

To further verify this, we washed the hematite nanoparticles with ethanol and water for more than 10 times, then calcinated them at 400 $^\circ\text{C}$ for 2 h. However, the additional washing and calcination treatments still cannot remove the C 1s signal. Referring to the literatures from different researchers, the residual organic carbon is

commonly detectable in XPS analysis. For example, the synthesized hematite {001}, {012}, {110} nanoparticles also presented clear C 1s XPS signal in the literatures⁴⁻⁶, demonstrating carbon contamination on hematite surface is difficult to avoid. Since the C compositions on the three hematite are very similar, the carbon contamination might stem from extra sources after synthesis. However, the discussion for the extra source of the carbon contamination is beyond the scope of this study, further investigation should be conducted.

Based on the above discussion, the collective XPS and isoelectric points results show that the three synthesized hematite are pure, with slight adventitious carbon contamination. However, the composition of C=O species is low (9-11%). Since only the C=O species can compete for the Lewis-acid sites, the C-C/C-H and C-N/C-O species have poor coordination capability, the slight carbon contamination should have little influence on the hydrolysis reaction. It would not challenge the main conclusion of current study.”

Figure R23. Zeta (ζ) potentials of HNP, HNR and HNC as a function of pH. The

isoelectric point is indicated by the point of $\zeta = 0$.

Figure R24. The N 1s (a, b) and C 1s (c-e) XPS of HNP, HNR and HNC. The contributions of different oxygen species for simulating O 1s were fitted according to Schöttner et al. (2019).

References:

- Chatman, S., Zarzycki, P. and Rosso, K.M. 2013. Surface potentials of (001), (012), (113) hematite (α -Fe₂O₃) crystal faces in aqueous solution. *Physical Chemistry Chemical Physics* 15(33), 13911-13921.
- Guo, T., Jiang, L., Wang, K., Li, Y., Huang, H., Wu, X. and Zhang, G. 2021. Efficient persulfate activation by hematite nanocrystals for degradation of organic pollutants under visible light irradiation: Facet-dependent catalytic performance and degradation mechanism. *Applied Catalysis B: Environmental* 286, 119883.
- Huang, X., Hou, X., Song, F., Zhao, J. and Zhang, L. 2016. Facet-dependent Cr(VI) adsorption of hematite nanocrystals. *Environmental Science & Technology* 50(4), 1964-1972.
- Li, T., Zhong, W., Jing, C., Li, X., Zhang, T., Jiang, C. and Chen, W. 2020. Enhanced hydrolysis of p-nitrophenyl phosphate by iron (hydr)oxide nanoparticles: Roles of exposed facets. *Environmental Science & Technology* 54(14), 8658-8667.
- Quast, K. 2016. The use of zeta potential to investigate the interaction of oleate on hematite. *Minerals Engineering* 85, 130-137.
- Schöttner, L., Ovcharenko, R., Nefedov, A., Voloshina, E., Wang, Y., Sauer, J. and Wöll, C. 2019. Interaction of water molecules with the α -Fe₂O₃(0001) surface: A combined experimental and computational study. *The Journal of Physical Chemistry C* 123(13), 8324-8335.
- Shen, Z., Zhang, Z., Li, T., Yao, Q., Zhang, T. and Chen, W. 2020. Facet-dependent adsorption

and fractionation of natural organic matter on crystalline metal oxide nanoparticles. *Environmental Science & Technology* 54(14), 8622-8631.

Yan, L., Chan, T. and Jing, C. 2020. Arsenic adsorption on hematite facets: spectroscopy and DFT study. *Environmental Science: Nano* 7(12), 3927-3939.

Li. 115. Has the presence of defects been investigated? They are expected to have a huge effect on reactivity.

Response: Thank you for the comment. In our previous studies, we developed a diffused reflectance infrared Fourier transform spectroscopy (DRIFTS) method to identify the Lewis-acid centers on iron mineral surface, using 2-chloro-N,N-dimethylacetamide (Cl-DMA) as the probe molecule (Wu et al., 2021), as Cl-DMA is volatile and its carbonyl group can complex with the exposed Fe sites. So, the Cl-DMA molecules could be purged into the DRIFTS system through a humidified N₂ flow. Inside the DRIFTS chamber (model HVC-DRP-5, Harrick Scientific, USA), the iron mineral particles were filled. The Cl-DMA molecules were cumulatively adsorbed by the exposed undercoordinated Fe sites on the mineral surface, then, the adsorption intensity and capacity could be recorded *in-situ* as indicated by the diffused reflectance IR signals. The varied interaction forces between surface and the carbonyl groups of Cl-DMA would induce different electron states of the carbonyl groups. Therefore, the surface Lewis-acidity can be precisely distinguished according to the extent of red-shift for $\nu(\text{C}=\text{O})$. We have used this method to probe the surface Lewis-acid sites on different iron minerals (ferrihydrite, goethite, hematite, maghemite) (Wu et al. 2021). As shown in Figure R25, the adsorbed Cl-DMA exhibits four distinct absorption peaks at $\sim 1643\text{ cm}^{-1}$, $1628\text{-}1630\text{ cm}^{-1}$, 1583 cm^{-1} and 1577 cm^{-1} , respectively. The corresponding shoulder peak at $\sim 1643\text{ cm}^{-1}$ on goethite represents for the hydrogen-bonding induced $\nu(\text{C}=\text{O})$ of Cl-DMA. While, the peak at $1628\text{-}1630\text{ cm}^{-1}$ on all the iron oxides indicates the octahedral Fe^{III} site complexed with Cl-DMA. Furthermore, the broad peak at 1583 cm^{-1} on maghemite is ascribed to the interaction from the tetrahedral Fe^{III} site, which is specific for maghemite. Finally, the less intensive peak at 1577 cm^{-1} is assigned to the Fe^{III} sites with O-defect, since the defective sites usually possess lower coordination

number and relatively stronger Lewis-acidity. This method can clearly show the distribution and relative abundance of surface Lewis-acid sites.

In this study, we applied the same method to measure the surface Lewis-acid sites, including the defective sites, on the three hematites (HNP, HNR and HNC). The hematites were pre-treated at 105 °C for overnight to mimic the condition before the kinetic experiment. As shown in Figure R26a-c, two significant absorption peaks appear at 1716-1721 cm^{-1} and 1621-1628 cm^{-1} , respectively. According to our previous investigation, the peak at 1716-1721 cm^{-1} represents the physically adsorbed or weakly interacted $\nu(\text{C}=\text{O})$. And the peak at 1621-1628 cm^{-1} corresponds to the $\nu(\text{C}=\text{O})$ coordinated with the octahedral Fe^{III} sites. The full width at half maximum (FWHM) for the peak at 1621-1628 cm^{-1} is relatively small (i.e., 45 cm^{-1} for HNP, 27 cm^{-1} for HNR, and 47 cm^{-1} for HNC), suggesting that the surface Lewis-acid sites are unitary. No significant absorption peaks or bands were observed in the wavenumber range lower than 1620 cm^{-1} , indicating that the surface defective sites are not abundant.

However, the $\nu(\text{C}=\text{O})$ on HNC appears at relatively low wavenumber position, i.e., 1621 cm^{-1} on HNC vs. 1625 cm^{-1} on HNR and 1628 cm^{-1} on HNP. This is inconsistent with the Bader charge analysis results, as the valence electrons (VEs) were calculated to be 6.29, 6.36-6.58 and 6.39 for the exposed Fe on HNC, HNR and HNP, respectively. The Fe cations with higher VEs usually indicate lower coordination state and stronger coordination ability. The possible explanation is that the HNC surface contains a certain degree of O-defects. Then, the Fe adjacent to or beyond the O vacancy can remain higher VE (6.57-6.70) (Figure R15).

The three hematites were further calcinated at 400 °C for 2 h. Thus, the surface -OH/H₂O can be removed, especially for those on the defective sites. As shown in Figure 26d-f, the absorption band of Cl-DMA on HNP/HNR/HNC is significantly broadened on HNP400 and HNC400, and a new shoulder peak appears at ~1580 cm^{-1} , suggesting that the O-defective sites are involved to adsorb Cl-DMA. By comparison, the HNR surface is more stoichiometric. In current study, the experimental mass loading of DMP/DnBP is far below the surface Fe sites, so DMP/DnBP molecules

prefer to interact/react with the O-defective Fe sites. However, the extent of O-defects on HNP and HNC is small.

In the revised manuscript, surface O-defects on HNC surface has been discussed thoroughly. Firstly, in the section “Determination of facet termination”, page 10-11, line 186-190: “For the {012} facet, the stoichiometric termination with arm-chair like topology ($O_3-Fe_5-O_4-Fe_6-O_4-R$, Fig. 3c) was calculated as the most stable surface ($\gamma = 1.95 \text{ J}\cdot\text{m}^{-2}$, Supplementary Table 1), which is consistent with the prior studies^{52,53}. Since its exposed Fe atoms are five-fold coordinated, the VE number is expectedly low (VE = 6.29, Supplementary Fig. 14d).”

In page 11-12, line 198-217: “It is important to note that the higher VEs of Fe cations usually implies lower coordination state and stronger coordination ability. However, when we further applied the diffused reflectance infrared Fourier transform spectroscopy (DRIFTS), using the compound 2-chloro-N,N-dimethylacetamide (Cl-DMA) as the probe molecule, to identify the surface Lewis-acid sites¹², the results are discrepant to the Bader charge analysis (Fig. 3, Supplementary Fig. 14). For example, HNC exhibits wider red-shift of $\nu_{C=O}$, which denotes to the carbonyl stretching vibration (1621 cm^{-1} on HNC vs. 1628 cm^{-1} on HNP & 1625 cm^{-1} on HNR, Supplementary Fig. 15a-c), suggesting that the HNC surface is more Lewis-acidic. This discrepancy might be ascribed to the existence of surface O-defects on actual HNC surface⁵³. By introducing a certain degree of O-defects on the surface of {012} facet model, the Fe atoms adjacent to the O-vacancies possess 6.70 - 6.74 VEs (Fig. 3c, Supplementary Fig. 14d), corresponding to the stronger Lewis-acidity. Meanwhile, a better fit for the IR spectra was also obtained using the {012} facet with O-defect for theoretical modelling (more discuss is shown below). Further study shows that the HNC400 (i.e., HNC calcinated at $400 \text{ }^\circ\text{C}$ for 2 h) exhibits a new shoulder peak at 1582 cm^{-1} (Supplementary Fig. 15f), probably stemming from the thermal desorption of -OH groups and H_2O molecules at the defective sites⁵⁶, suggesting a small amount of surface defective sites. For comparison, the $\nu_{C=O}$ peaks in both HNR and HNR400 samples are narrow and sharp (Supplementary Fig. 15b, e), indicating negligible surface defects on HNR.”

Secondly, the theoretical IR spectra of the DMP- $\{012\}$ slab model indicate that DMP coordinated with Fe on the defective site can induce wider red-shift of $\nu(\text{C}=\text{O})$ to the wavenumber $< 1600 \text{ cm}^{-1}$. In the revised manuscript, the related information is added in page 16, line 308-315: “For the $\{012\}$ facet without O-defect, the $\nu_{\text{C}=\text{O}}$ of *cis*-DMP shows distinct absorption peaks at $1645 (\nu_1) / 1625 (\nu_2) \text{ cm}^{-1}$ or $1674 (\nu_1) / 1607 (\nu_2) \text{ cm}^{-1}$ (Supplementary Fig. 18e-f). While, on the O-defective site of $\{012\}$, the $\nu_{\text{C}=\text{O}}$ appears at $1627 (\nu_1)$ and $1590 (\nu_2) \text{ cm}^{-1}$ (Fig. 4c₃). Therefore, the theoretical IR spectra of DMP on the slab models directly show the coordination modes. In general, bidentate coordination can induce stronger Lewis-acid interaction than monodentate coordination, imposing wider red-shift of $\nu_{\text{C}=\text{O}}$ even to $< 1600 \text{ nm}^{-1}$.”

Thirdly, the additional kinetic experiments of DMP on HNC in a wide range of surface concentrations also provide another evidence that the HNC surface contains a certain degree of defects ($0.07 \text{ sites}\cdot\text{nm}^{-2}$), page 20, line 397-403: “To further verify this hypothesis, we examined the concentration effect ($[\text{DMP}]_0 = 2 - 20 \mu\text{mol}\cdot\text{g}^{-1}$) on the reaction. As shown in Supplementary Fig. 25, the hydrolysis rates of DMP on HNC decrease greatly as the initial DMP concentration exceeds $2 \mu\text{mol}\cdot\text{g}^{-1}$, suggesting that the O-vacancy density on HNC surface might be $\sim 2 \mu\text{mol}\cdot\text{g}^{-1}$ (i.e., $0.07 \text{ sites}\cdot\text{nm}^{-2}$), accounting for $\sim 2\%$ of the total surface area. By comparison, the hydrolysis rates of DMP on HNR only slightly decreased in the applied concentration range, due to its stoichiometric surface condition.”

Figure R25. *In-situ* DRIFT spectroscopic measurements of the adsorbed 2-chloro-N,N-dimethylacetamide (Cl-DMA) on (a) ferrihydrite, (b) goethite, (c) hematite, (d) maghemite. The lines in different colors represent the cumulative adsorption of Cl-DMA during gas dosing (0-490 min) of 10 min interval. This figure was cited from a reference (Wu et al., 2021).

Figure R26. *In-situ* DRIFTS measurements for surface Lewis-acid sites. The 2-chloro-N,N-dimethylacetamide (Cl-DMA) was used as the probe molecule. Cl-DMA was cumulatively adsorbed by the surface active sites on HNP (a), HNR (b), HNC (c), HNP400 (d), HNR400 (e) and HNC400 (f). The hematite of (d-f) were pre-calcinated at 400 °C for 2 h. The cumulative adsorption of Cl-DMA was recorded in 10 min interval. The peaks in the red range indicate the Lewis-acid interacted Cl-DMA. The peaks in the yellow range indicate the physical adsorbed Cl-DMA. The reversal peaks in the blue range indicate that the surface bonded -OH groups and chemisorbed H₂O molecules are gradually substituted by Cl-DMA.

Figure R15. Bader charge analysis of the surface exposed Fe atoms on the {001} facet with one O vacancy (a), the {104} facet-layer 5 (b), the {012} facet-layer 1 with one O vacancy (c). VE means the valence electrons remaining on Fe.

Reference:

Wu, D., Huang, S., Zhang, X., Ren, H., Jin, X. and Gu, C. 2021. Iron minerals mediated interfacial hydrolysis of chloramphenicol antibiotic under limited moisture conditions. *Environmental Science & Technology* 55(14), 9569-9578.

Li 121. The number of digits for the SSAs should be decreased (18.1 instead of 18.08 for example).

Response: Thank you for the correction. In the revised manuscript, the number of digits has been modified.

Li 141. The term “water content of 400%” is troublesome, as “over-wet condition”. If the authors mean that liquid water is present, they could give the calculated value of water layers at the surface of the particles for example.

Response: Thank you for the comment. The term “water content of 400%” means that in the reaction, 50 mg hematite nanoparticles were mixed with 200 μL water, which represents the reaction occurring under the aqueous-like condition. It was reported that, on $\alpha\text{-Fe}_2\text{O}_3$ (001), the coverage of water reaches 1 monolayer at $\sim 15\%$ RH, and

increases to 1.5 monolayer at 34% RH (Yamamoto et al., 2010). Obviously, under the condition (50 mg nanoparticles in 200 μ L water), the water content of hematite is oversaturated. As shown in Figure R27, the hematite particles are immersed into the water. To avoid ambiguity, in the revised manuscript, the expression “water content of 400%” has been modified to “**water oversaturated condition**”.

Figure R27. The hematite sample under RH = 76% (left) and water oversaturated condition (right, 50 mg/L nanoparticles in 200 μ L pure water).

Reference:

Yamamoto, S., Kendelewicz, T., Newberg, J.T., Ketteler, G., Starr, D.E., Mysak, E.R., Andersson, K.J., Ogasawara, H., Bluhm, H., Salmeron, M., Brown, G.E. and Nilsson, A. 2010. Water adsorption on α -Fe₂O₃(0001) at near ambient conditions. *The Journal of Physical Chemistry C* 114(5), 2256-2266.

Li. 165. Uncoordinated Fe sites are expected to be very reactive towards water molecules as a vapour or in a liquid. Thus, this type of sites may be not exist in the conditions of the synthesis (in water or at RH > 0%). This type of question arises in the surface chemistry of transition aluminas, and the authors should give some evidence (experimental or from calculations) that the terminations they propose (li.161-163) actually exist in their systems.

Response: Thank you for the comments. It is true that the surface exposed undercoordinated Fe sites are reactive to water molecules, leading to surface hydroxylation and hydration, which are complicated and vary with respect to the exposed water partial pressure (Yamamoto et al., 2010). Liu et al. (1998) applied XPS analysis to investigate the reaction between water vapor and clean {001} surface of α -Al₂O₃ and α -Fe₂O₃, and found that water molecules would be mainly dissociatively adsorbed at the defective sites below the threshold water partial pressure (i.e., ~1 Torr for α -Al₂O₃, and ~0.0001 Torr for α -Fe₂O₃). The dissociated OH⁻ group would attach to the surface Fe³⁺ ion and the proton to the surface O²⁻ ion (Nguyen et al., 2013). Above the threshold pressure, extensive hydroxylation would occur within one monolayer (Liu et al., 1998). Further increase of water partial pressure would induce more water layers with interactive hydrogen bonding to surface coordinated -OH groups and structural O, namely the hydration layer. By the way, the RH 76% corresponds to the water partial pressure of ~18 Torr, there should be more than two water layers on hematite surface (Yamamoto et al., 2010). Trainor et al. (2004) proposed that the terminal Fe of hematite {001} would bond with three -OH groups to form the (HO)₃-Fe-O₃-R termination. The hematite {012} surface was reported to be more reactive with water molecules than the {001} surface (Voloshina, 2018), and there are three possible types of (hydr)oxo functional groups exposed on the surface: Fe-(OH)₂, Fe-OH and Fe₂-(OH)₂ (Tanwar et al., 2007).

The surface hydroxylation and hydration would significantly affect the reactivity of the hematite, as the hydroxyl groups and the chemisorbed water molecules would compete with the target compounds for the reactive sites. The competition effect has been examined in our previous studies, that the hydrolysis rates of chloramphenicol antibiotic on iron oxides greatly decreased as the increase of water partial pressure (RH > 76%) (Wu et al., 2021). Likewise, in current study, the hydrolysis rate of DMP/DnBP in water is about 2-orders of magnitude slower than that under RH 76% by the hematite.

However, surface hydroxylation/hydration would not significantly change the reaction mechanism, since DMP/DnBP molecules mainly coordinate with the exposed

Fe site instead of surface oxygen or hydroxyl groups. It could be evidenced by the *in-situ* DRIFTS measurement of DMP on HNP/HNR/HNC. As shown in Figure R28, the clear reversal peaks at 3000 - 3750 cm^{-1} represent the desorption of chemisorbed -OH/H₂O, substituted by DMP. The PAEs can compete with surface water molecules and hydroxyl groups for the Fe sites. Therefore, to simplify the facet model, surface hydroxylation/hydration was not considered for modelling. Actually, many previous studies also used the bare Fe terminal slabs for modelling, without considering surface hydroxylation/hydration, when investigating the interaction between iron and organic compounds (Cao et al., 2019; Han et al., 2021; Huang et al., 2019; Huang et al., 2017; Wu et al., 2020; Zhou et al., 2012).

In order to determine the facet surface termination and surface condition, we firstly calculated the surface energy for different terminations of each facet. The termination with the lowest surface energy is regarded as the most stable termination, i.e., the {001}-layer 1, the {104}-layer 1, and the {012}-layer 1 termination (Figure R29). Secondly, the proper hematite slabs with a vacuum space of 15 Å were applied for stimulating the adsorption configurations of DMP on HNP/HNR/HNC. Thirdly, the adsorption configurations of DMP on the {001}-layer 1, {104}-layer 1 & layer 5, {012}-layer 1 slabs were subjected for vibration analysis of DMP. In comparison with our experimental IR results, the surface condition can be further corrected. For the stoichiometric {001}-layer 1 termination, the adsorbed DMP results in the $\nu(\text{C}=\text{O})$ at 1640 cm^{-1} , which is slightly higher than the experimental observation (1621 cm^{-1}). Therefore, the actual {001}-layer 1 termination should contain a certain degree of O-defects, which can increase the Lewis-acidity of the exposed Fe. Likewise, for the stoichiometric {012}-layer 1 termination, the adsorbed DMP presents the $\nu(\text{C}=\text{O})$ at 1645 (ν_1) / 1625 (ν_2) cm^{-1} and 1674 (ν_1) / 1607 (ν_2) cm^{-1} (Figure R30). While, DMP adsorbed on the O-defective site shows the $\nu(\text{C}=\text{O})$ at 1627 (ν_1) and 1590 (ν_2) cm^{-1} (Figure R30), which are much closer to the experimental results (Figure R31). With the involvement of O vacancy, a better fit was obtained for the calculated IR spectra compared to our experimental results. While, the IR spectra of DMP adsorbed on the

{104}-layer 1 and layer 5 can fit well with the experimental IR data (Please see the revised manuscript, Figure 4 and Supplementary Figure 18). Therefore, we can conclude that the actual {001} and {012} facets contain a certain degree of O-defects, however, the {104} facet has negligible defects.

Although the surface hydroxylation/hydration was not considered for modeling, the effect of hydration and hydroxylation on surface reaction was also discussed in the revised manuscript. The remaining valence electrons (VEs) of the exposed Fe atom can be used to indicate its catalytic activity. For {001}-layer 1, the VEs of the exposed Fe atoms are moderate even they are three-fold coordinated (Figure R32a). With one O-defect on {001} facet, the average VEs of the exposed Fe are increased. While, with the surface hydroxylation, the VEs of the exposed Fe are reduced (Figure R32a). For the {104}-layer 1, the dissociative bonding with a low extent of H₂O molecules adventitiously reduces the VEs of the adjacent Fe_{subsurface}, and increases the VEs of the adjacent Fe_{outmost} (Figure R32b). Furthermore, for {012} facet, the presence of O-defect increases the average VEs of the exposed Fe and more defects would contribute to the higher VEs of the Fe (Figure R32d). Additionally, dissociative bonding with a low extent of H₂O molecules can also induce surface charge redistribution (Figure R32d).

Based on above discussion, we have carefully proposed the terminations of the facets. In the revised manuscript, we add one paragraph to discuss this aspect, in page 10-13, line 172-235: “**Determination of facet terminations.** The catalytic activity of hematite typically depends on its surface properties. However, the {001}, {104}, {012} facets have multiple theoretical surface terminations (Supplementary Fig. 11 - Fig. 13). To determine the exact termination, we calculated the surface energies of different possible terminations for each facet. It is generally accepted that the facet with the lowest surface energy is thermodynamically favorable. For the {001} facet, after surface relaxation, its layer 1 termination (Fe₃-O₃-Fe₆-R, the subscript represents for the coordination number, Fig. 3a) possesses much lower surface energy ($\gamma = 1.86 \text{ J}\cdot\text{m}^{-2}$) than the other terminations (Supplementary Table 1), which is in accordance with the previous reports⁴⁹⁻⁵¹. The exposed Fe atoms on {001}-layer 1 termination are three-fold

coordinated. However, according to Bader charge analysis, the valence electrons (VEs) remaining on the exposed Fe are relatively low (VEs = 6.39, indicating 1.61 VEs are transferred from Fe to the adjacent O) (Fig. 3a), since the exposed Fe atoms are relaxed inward to the underlayer plane of O atoms with significant charge redistribution.

For the {012} facet, the stoichiometric termination with arm-chair like topology ($\text{O}_3\text{-Fe}_5\text{-O}_4\text{-Fe}_6\text{-O}_4\text{-R}$, Fig. 3c) was calculated as the most stable surface ($\gamma = 1.95 \text{ J}\cdot\text{m}^{-2}$, Supplementary Table 1), which is consistent with the prior studies^{52,53}. Since its exposed Fe atoms are five-fold coordinated, the VE number is expectedly low (VE = 6.29, Supplementary Fig. 14d).

Compared to the other two facets, {104} facet was less investigated^{54,55}. In this study, the {104}-layer 1 termination ($\text{O}_3\text{-Fe}_4\text{-Fe}_4\text{-O}_4\text{-O}_4\text{-R}$, Fig. 3b) is calculated with the lowest surface energy ($\gamma = 1.94 \text{ J}\cdot\text{m}^{-2}$), following with the {104}-layer 5 termination ($\text{O}_2\text{-O}_3\text{-Fe}_5\text{-Fe}_6\text{-O}_3\text{-R}$, $\gamma = 2.12 \text{ J}\cdot\text{m}^{-2}$) (Supplementary Fig. 12). As both terminations possess the relatively low γ , they may co-exist with the domain of {104}-layer 1 as the stable {104} termination. The calculated VEs for the subsurface Fe (6.58) are higher than the outmost Fe (6.36) on the {104}-layer 1 (Fig. 3b).

It is important to note that the higher VEs of Fe cations usually implies lower coordination state and stronger coordination ability. However, when we further applied the diffused reflectance infrared Fourier transform spectroscopy (DRIFTS), using the compound 2-chloro-N,N-dimethylacetamide (Cl-DMA) as the probe molecule, to identify the surface Lewis-acid sites¹², the results are discrepant to the Bader charge analysis (Fig. 3, Supplementary Fig. 14). For example, HNC exhibits wider red-shift of $\nu_{\text{C=O}}$, which denotes to the carbonyl stretching vibration (1621 cm^{-1} on HNC vs. 1628 cm^{-1} on HNP & 1625 cm^{-1} on HNR, Supplementary Fig. 15a-c), suggesting that the HNC surface is more Lewis-acidic. This discrepancy might be ascribed to the existence of surface O-defects on actual HNC surface⁵³. By introducing a certain degree of O-defects on the surface of {012} facet model, the Fe atoms adjacent to the O-vacancies possess 6.70 - 6.74 VEs (Fig. 3c, Supplementary Fig. 14d), corresponding to the stronger Lewis-acidity. Meanwhile, a better fit for the IR spectra was also obtained

using the {012} facet with O-defect for theoretical modelling (more discuss is shown below). Further study shows that the HNC400 (i.e., HNC calcinated at 400 °C for 2 h) exhibits a new shoulder peak at 1582 cm⁻¹ (Supplementary Fig. 15f), probably stemming from the thermal desorption of -OH groups and H₂O molecules at the defective sites⁵⁶, suggesting a small amount of surface defective sites. For comparison, the ν_{C=O} peaks in both HNR and HNR400 samples are narrow and sharp (Supplementary Fig. 15b, e), indicating negligible surface defects on HNR.

Moreover, the surface undercoordinated Fe sites are reactive with water molecules. Under low water partial pressure, H₂O molecules would entirely or dissociatively bond to lattice Fe and O with interactive hydrogen bonding, leading to surface hydroxylation, hydration and protonation (Fig. 3, Supplementary Fig. 14)⁵⁷⁻⁶⁰. The O 1s signals in XPS spectra demonstrate that the HNP, HNR and HNC surfaces are partially hydroxylated and hydrated (bond -OH: 16-21.3%, chemisorbed H₂O: 13.0 - 17.3%) even under vacuum condition (Supplementary Fig. 4 - Fig. 6). Therefore, under ambient condition, their surface should be more hydroxylated and hydrated⁶¹. Although surface hydroxylation/hydration could shield the surface reaction, and affect the valency of the exposed Fe, organic ligands can still compete with the surface bond -OH/H₂O for the undercoordinated Fe sites, especially under low moisture condition⁶², which is strongly evidenced by the desorption of the chemisorbed -OH/H₂O (3000 - 3750 cm⁻¹) substituted by either Cl-DMA or DMP on the three facets (Supplementary Fig. 15 & 16). Therefore, the surface hydroxylation/hydration would not change the coordination mode and the catalytic mechanism. To simplify the facet model, surface hydroxylation/hydration was not involved for modelling. In current study, the {001}-layer 1, {104}-layer 1 co-existing with layer 5, and the {012}-layer 1 with O-defects are deduced as the surface terminations for HNP, HNR and HNC, respectively (Fig. 3).”

Figure R28. *In-situ* DRIFTS measurements by applying gaseous DMP. The blue range demonstrates the desorption of the surface bonded -OH groups and chemisorbed H₂O molecules, substituted by gaseous DMP.

Figure R29. The stable surface terminations of the hematite {001}, {104} and {012} facets before and after surface relaxation.

Figure R30. The adsorption configurations of DMP on the {012} facet with/without one O-defect, as well as the corresponding IR spectra.

Figure R31. The experimental IR spectra of DMP before and after adsorbing onto HNP, HNR and HNC, the *ex-situ* KBr wafer method (left), and the *in-situ* DRIFTS of DMP adsorbed onto HNP, HNR and HNC under RH 76% (right).

Figure R32. Bader charge analysis of the facet-exposed Fe. The valence electron (VE) numbers on the outmost Fe atoms are marked in light yellow, and those on the subsurface Fe atoms are marked in sky blue. The green disks indicate the O vacancies, and the yellow circles highlight the hydroxylation and hydration positions.

References:

- Cao, S., Zhang, X., Huang, X., Wan, S., An, X., Jia, F. and Zhang, L. 2019. Insights into the facet-dependent adsorption of phenylarsonic acid on hematite nanocrystals. *Environmental Science: Nano* 6(11), 3280-3291.
- Han, R., Lv, J., Zhang, S. and Zhang, S. 2021. Hematite facet-mediated microbial dissimilatory iron reduction and production of reactive oxygen species during aerobic oxidation. *Water Research* 195, 116988.
- Huang, X., Chen, Y., Walter, E., Zong, M., Wang, Y., Zhang, X., Qafoku, O., Wang, Z. and Rosso, K.M. 2019. Facet-specific photocatalytic degradation of organics by heterogeneous fenton chemistry on hematite nanoparticles. *Environmental Science & Technology* 53(17), 10197-10207.
- Huang, X., Hou, X., Song, F., Zhao, J. and Zhang, L. 2017. Ascorbate induced facet dependent reductive dissolution of hematite nanocrystals. *Journal of Physical Chemistry C* 121, 1121.
- Liu, P., Kendelewicz, T., Brown, G.E., Nelson, E.J. and Chambers, S.A. 1998. Reaction of water vapor with α -Al₂O₃(0001) and α -Fe₂O₃(0001) surfaces: synchrotron X-ray photoemission studies and thermodynamic calculations. *Surface Science* 417(1), 53-65.
- Nguyen, M.T., Seriani, N. and Gebauer, R. 2013. Water adsorption and dissociation on α -Fe₂O₃(0001): PBE+U calculations. *The Journal of Chemical Physics* 138(19), 194709.
- Tanwar, K.S., Lo, C.S., Eng, P.J., Catalano, J.G., Walko, D.A., Brown, G.E., Waychunas, G.A., Chaka, A.M. and Trainor, T.P. 2007. Surface diffraction study of the hydrated hematite (11 $\bar{0}2$) surface. *Surface Science* 601(2), 460-474.
- Trainor, T.P., Chaka, A.M., Eng, P.J., Newville, M., Waychunas, G.A., Catalano, J.G. and Brown, G.E.J. 2004. Structure and reactivity of the hydrated hematite (0001) surface. *Surface Science* 573, 204-224.
- Voloshina, E. (2018) *Encyclopedia of Interfacial Chemistry*. Wandelt, K. (ed), pp. 115-121, Elsevier, Oxford.
- Wu, D., Huang, S., Zhang, X., Ren, H., Jin, X. and Gu, C. 2021. Iron minerals mediated interfacial hydrolysis of chloramphenicol antibiotic under limited moisture conditions. *Environmental Science & Technology* 55(14), 9569-9578.
- Wu, S., He, Y., Wang, C., Zhu, C., Shi, J., Chen, Z., Wan, Y., Hao, F., Xiong, W., Liu, P. and Luo, H. 2020. Selective Cl-decoration on nanocrystal facets of hematite for high-Efficiency catalytic oxidation of cyclohexane: Identification of the newly formed Cl-O as active sites. *ACS Applied Materials & Interfaces* 12(23), 26733-26745.
- Yamamoto, S., Kendelewicz, T., Newberg, J.T., Ketteler, G., Starr, D.E., Mysak, E.R., Andersson, K.J., Ogasawara, H., Bluhm, H., Salmeron, M., Brown, G.E. and Nilsson, A. 2010.

Water adsorption on $\alpha\text{-Fe}_2\text{O}_3(0001)$ at near ambient conditions. *The Journal of Physical Chemistry C* 114(5), 2256-2266.

Zhou, X., Lan, J., Liu, G., Deng, K., Yang, Y., Nie, G., Yu, J. and Zhi, L. 2012. Facet-mediated photodegradation of organic dye over hematite architectures by visible light. *Angewandte Chemie International Edition* 51(1), 178-182.

TOC. The graph seems cut. Moreover, there are too many details, and it is difficult to understand in an independent way.

Response: Thank you for the comment. In the revised manuscript, the new TOC graph has been provided.

Fig. 4. "neutral"

Response: Thank you for the correction. In the revised manuscript, it has been corrected as Figure 5.

REVIEWERS' COMMENTS

Reviewer #2 (Remarks to the Author):

The authors made a major revision and carefully addressed all of my questions. In my opinion, the manuscript can be accepted in its current form.

Reviewer #3 (Remarks to the Author):

The authors answered all the questions, have done complementary analysis, and have completed the manuscript according to the recommendations.

I support the publication of this work.

Response to Reviewers' Comments

Reviewer #2 (Remarks to the Author):

The authors made a major revision and carefully addressed all of my questions. In my opinion, the manuscript can be accepted in its current form.

Response: We thank the reviewer again for his/her valuable suggestions and affirmation of our revised manuscript.

Reviewer #3 (Remarks to the Author):

The authors answered all the questions, have done complementary analysis, and have completed the manuscript according to the recommendations.

I support the publication of this work.

Response: We thank the reviewer again for his/her valuable suggestions and affirmation of our revised manuscript.